# IVEBench: Modern Benchmark Suite for Instruction-Guided Video Editing Assessment

**Yinan Chen**[1]*, **Jiangning Zhang**[1,2]*, **Teng Hu**[3], **Yuxiang Zeng**[4], **Zhucun Xue**[1],
**Qingdong He**[2], **Chengjie Wang**[2,3], **Yong Liu**[1]†, **Xiaobin Hu**[2], **Shuicheng Yan**[5]
[1]Zhejiang University,  [2]Tencent Youtu Lab,  [3]Shanghai Jiao Tong University
[4]University of Auckland,  [5]National University of Singapore

## Abstract

Instruction-guided video editing has emerged as a rapidly advancing research direction, offering new opportunities for intuitive content transformation while also posing significant challenges for systematic evaluation. Existing video editing benchmarks fail to support the evaluation of instruction-guided video editing adequately and further suffer from *limited source diversity, narrow task coverage and incomplete evaluation metrics*. To address the above limitations, we introduce IVEBench, a modern benchmark suite specifically designed for instruction-guided video editing assessment. IVEBench comprises a diverse database of 600 high-quality source videos, spanning seven semantic dimensions, and covering video lengths ranging from 32 to 1,024 frames. It further includes 8 categories of editing tasks with 35 subcategories, whose prompts are generated and refined through large language models and expert review. Crucially, IVEBench establishes a three-dimensional evaluation protocol encompassing video quality, instruction compliance and video fidelity, integrating both traditional metrics and multimodal large language model-based assessments. Extensive experiments demonstrate the effectiveness of IVEBench in benchmarking state-of-the-art instruction-guided video editing methods, showing its ability to provide comprehensive and human-aligned evaluation outcomes. All data and code will be made publicly available.

## 1 Introduction

Video editing, which aims to transform source videos to satisfy user-specified editing requirements, has emerged as a crucial capability in both creative industries and practical applications. As the field of generative modeling and multimodal understanding advances (Yang et al., 2024; Bai et al., 2025b), Instruction-guided Video Editing (abbreviated as IVE that edits are directed by natural language instruction) has attracted significant research interest (Cheng et al., 2023). This paradigm promises intuitive and fine-grained control over video content, unlocking new possibilities for content creation, entertainment, and human-computer interaction.

Despite rapid progress, current video editing benchmarks still present notable limitations. Existing benchmarks (Sun et al., 2025b) suffer from three major challenges: *i)* **Insufficient diversity in video sources**: The coverage of semantic categories, scenes, and editing instructions remains limited, constraining the generalizability of evaluation results (Chen et al., 2025; Li et al., 2025b). *ii)* **Restricted editing prompts**: Editing instructions are often narrowly defined or lack granularity, failing to reflect the diverse and complex requirements of real-world editing scenarios (Wang et al., 2025a). *iii)* **Fragile evaluation metrics**: Current evaluation protocols are frequently restricted to basic quality or alignment measures, lacking a comprehensive, multidimensional assessment, especially those leveraging advances in Multimodal Large Language Models (MLLMs) for semantic understanding (Sun et al., 2025b; Chen et al., 2025). Besides, existing benchmarks are primarily designed for video editing methods based on source-target prompts. *However, due to their poor user-friendliness and unclear editing requirements, mainstream video editing methods have now*

---

*Equal contribution.

†Corresponding author.

*shifted toward instruction-guided approaches (Cheng et al., 2023), mirroring a similar trend in image editing (Brooks et al., 2023).* Therefore, **there is an urgent need for a comprehensive benchmark that fully supports IVE.**

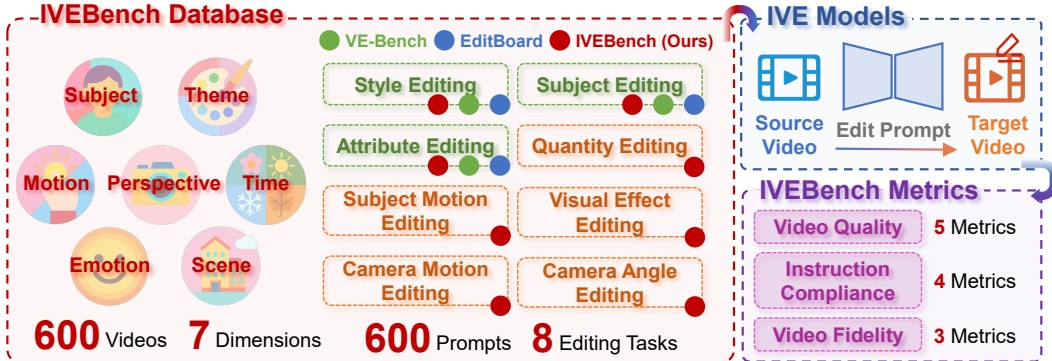

Figure 1: **Overview of our proposed IVEBench**. *1)* We construct a diverse video corpus consisting of 600 high-quality source videos systematically organized across 7 semantic dimensions. *2)* For source videos, we design carefully crafted edit prompts, covering 8 major editing task categories with 35 subcategories. *3)* We establish a comprehensive three-dimensional evaluation protocol comprising 12 metrics, enabling human-aligned benchmarking of state-of-the-art IVE methods.

In this paper, we propose a modern benchmark suite termed **IVEBench** for IVE assessment, which tackles the aforementioned challenges through three key innovations: *1)* **Diverse video corpus**: We construct a highly diverse dataset of 600 source videos, systematically collected and filtered to cover a wide range of topics across 7 semantic dimensions (see Fig. 1). *2)* **Comprehensive editing prompts**: Editing tasks are designed to cover 8 categories, with prompts generated and refined via Large Language Models (LLMs) and expert review. *3)* **Robust evaluation metrics**: We introduce a three-dimensional evaluation protocol encompassing video quality, instruction compliance, and video fidelity, incorporating both traditional metrics and MLLM-based assessments for richer, more objective evaluation.

We systematically demonstrate that our evaluation suite exhibits a high degree of alignment with human perception across all metrics. Through both qualitative and quantitative analyses of mainstream IVE methods, we provide valuable insights for the field of video editing. We will open-source the code, release the dataset, and keep track of the latest IVE methods.

## 2 RELATED WORK

**Instruction-guided video editing.** In recent years, the rapid advancement of image editing technologies has laid a solid foundation for video editing tasks. As the demand for understanding and generating higher-dimensional content increases, research focus has gradually shifted from static image editing to dynamic video editing (Wu et al., 2023). Early video editing methods are initially influenced by inversion techniques in the image editing domain (mainly DDIM Inversion (Song et al., 2021)), leading to the development of numerous source-target prompt-based editing approaches (Qi et al., 2023; Ceylan et al., 2023; Li et al., 2024). Although these approaches can accurately preserve object locations and poses during the inversion process (Geyer et al., 2023; Jeong & Ye, 2023; Cong et al., 2023), they are inherently limited when it comes to editing tasks involving subject movement or camera motion (Yatim et al., 2024; Kara et al., 2024). Furthermore, rather than providing detailed target prompts, users tend to express their editing requirements through instructions (Cheng et al., 2023). Given these limitations, IVE methods have gained burgeoning attention in the industry due to their greater user-friendliness and adaptability to diverse editing needs (Cheng et al., 2023). Mainstream approaches typically combine InstructPix2Pix (Brooks et al., 2023) for first-frame editing and then leverage generative models to propagate the modifications across the entire video (Khachatryan et al., 2023; Ku et al., 2024; Liu et al., 2024a). To overcome the quality bottleneck in such pipelines, Ditto (Bai et al., 2025a) further introduces a scalable synthetic data generation pipeline, leveraging high-quality image editors and in-context video generators to construct massive high-fidelity editing pairs. In contrast to these paradigms, InsV2V (Cheng et al., 2023) retrains the model on synthetic triplets of input video, editing instruction, and target video, enabling direct learning of instruction-

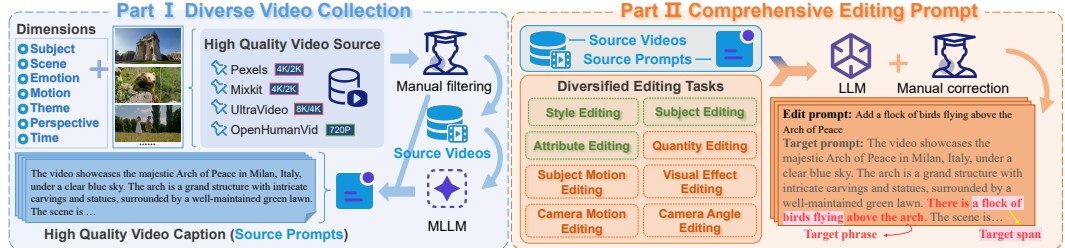

Figure 2: Data acquisition and processing pipeline of **IVEBench** includes: *1)* Curation process to 600 high-quality diverse videos. *2)* Well-designed pipeline for comprehensive editing prompts.

driven video modification for consistent long video editing. Building on this, InsViE-1M (Wu et al., 2025) further adopts multi-stage training on CogVideoX-2B (Yang et al., 2024) and supports static video editing tasks involving camera motion. Lucy-Edit (Team, 2025) employs a rectified-flow framework with channel concatenation to achieve high-fidelity editing without requiring masks or auxiliary inputs. Subsequently, ICVE (Liao et al., 2025) introduces a low-cost pretraining strategy leveraging in-context learning from unpaired video clips to learn general editing concepts. Recently, there is a growing trend towards unifying video understanding, generation, and editing within a single framework by integrating MLLMs with Diffusion Transformers (DiTs). Omni-Video (Tan et al., 2025) and UniVideo (Wei et al., 2025) propose dual-stream or unified architectures where the MLLM handles complex multimodal instruction understanding and guides the DiT for consistent video generation and editing. Based on this, InstructX (Mou et al., 2025) further utilizing mixed image-video training to transfer robust editing capabilities to the video domain.

**Video editing benchmarks.** With the introduction of benchmarks such as VBench (Huang et al., 2024) and T2V-CompBench (Sun et al., 2025a), the evaluation systems in the field of video generation have become increasingly comprehensive. Concurrently, video editing has also garnered significant attention, leading to the recent emergence of dedicated benchmarks for text-driven video editing. Among them, VE-Bench (Sun et al., 2025b) and EditBoard (Chen et al., 2025) introduce dedicated datasets and evaluation systems for text-driven video editing, partially covering editing tasks of subject, style and attribute editing. Building on these foundations, FiVE (Li et al., 2025b) and TDVE-Assessor (Wang et al., 2025a) further push evaluation by proposing MLLM-based metrics that enhance the objectivity of evaluation. However, a significant limitation of these existing benchmarks is that they are designed to support source-target prompt-based editing methods, while offering no or only partial support for IVE methods (Sun et al., 2025b; Chen et al., 2025). Furthermore, these benchmarks are constrained by limited dataset sizes, narrow content coverage, and include only a small subset of editing task types (Li et al., 2025b; Wang et al., 2025a). To address these issues, we propose IVEBench, a thorough benchmark suite specifically designed for IVE methods.

## 3 IVEBENCH DATABASE

### 3.1 DIVERSE VIDEO COLLECTION FOR IVE

**Video data source.** To ensure the comprehensiveness of our benchmark for video editing evaluation, we first expand the semantic coverage of source videos. Specifically, we define seven semantic dimensions and further subdivide each dimension into multiple fine-grained topics, resulting in a total of 30 topics. These subdivisions form a diverse set of semantic requirements for source videos, as illustrated in Fig. 3 (b). Based on these requirements, we manually collect high-quality video samples (≥2K) on Pexels (Ingo et al., 2014) and Mixkit (Ta'eed & Assi, 2019), as well as some from open-source UltraVideo (Xue et al., 2025). In addition, we incorporate a subset from OpenHumanVid (Li et al., 2025a) dataset to further enhance the quantity and diversity of human-centric videos (see Fig. 2).

**Hybrid automated and manual filtering.** All candidate videos undergo a two-stage processing pipeline. In the automatic preprocessing stage, black borders, subtitles, and low-quality content are removed. Subsequently, during the manual screening stage, we further ensure that the video content is suitable for editing and capable of covering a wide spectrum of tasks ranging from simple to complex. Ultimately, we construct a source video dataset comprising 600 videos with comprehensive semantic coverage, high resolution, and varied frame lengths. The dataset is organized into two

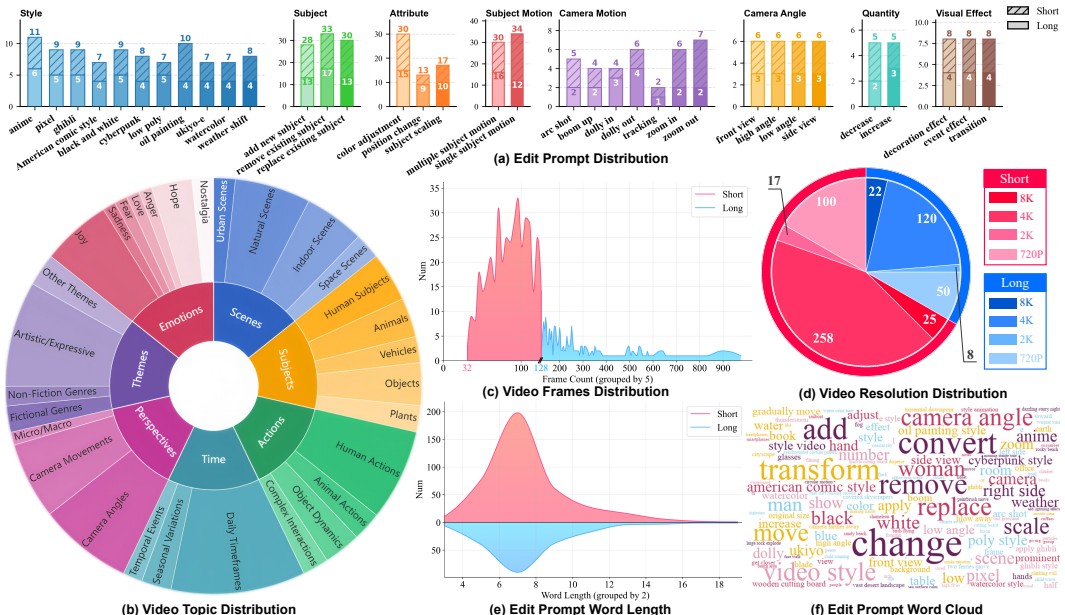

Figure 3: Statistical distributions of IVEBench.

subsets according to frame count: *i)* the short subset contains 400 videos ranging from 32 to 128 frames. *ii)* the long subset includes 200 videos ranging from 129 to 1,024 frames, representing a higher standard for long-sequence evaluation.

**Structural video caption.** After obtaining the high-quality source videos, we employ Qwen2.5-VL-72B (Bai et al., 2025b) to generate captions of appropriate length for each video, capturing key aspects such as subjects, backgrounds, subject actions, emotional atmosphere, visual styles, as well as camera perspectives and movements. These annotated attributes are designed to form a structured vocabulary of editable elements, establishing a robust foundation for subsequent user-driven modification requests.

## 3.2 Comprehensive IVE Prompt Generation

**Diversified editing objectives.** To ensure comprehensive coverage of task types in our benchmark for video editing evaluation, we categorize the editing prompts into eight major classes. Each of these main categories is further subdivided into more fine-grained subcategories, resulting in a total of 35 subcategories, as illustrated in Fig. 3 (a). Together, these eight categories encompass the full range of current requirements for IVE tasks and effectively address the limitations of existing benchmarks in terms of task coverage.

**LLM-assisted prompt generation and selection.** For each source video, we employ Doubao-1.5-pro (Seed et al., 2025), together with previously obtained detailed captions, to automatically select the most suitable editing category and generate a corresponding editing prompt. In addition, the system simultaneously produces the associated target prompt and target phrase, which serve as references for subsequent evaluation metrics. This design ensures that our benchmark can also accommodate text-driven video editing methods. All editing categories and prompts are further manually reviewed and refined to guarantee balanced category distribution as well as clear and reasonable prompts.

## 4 Comprehensive Metrics of IVEBench

In the context of IVE tasks, we define a video editing instance as comprising three data elements: the source video, the edit prompt (*i.e.*, the editing instruction expressed in natural language), and the target video. Based on the relationships among these elements, we evaluate the target video along three dimensions: *1)* **Video Quality** focuses on the quality of the target video itself; *2)* **Instruction Compliance** focuses on the alignment between the edit prompt and the target video; *3)* **Fidelity** focuses on the consistency between the source video and the target video. Notably, the dimensions

of Video Quality and Instruction Compliance are also applicable as evaluation criteria in video generation tasks, whereas Fidelity is a dimension specific to video editing.

## 4.1 VIDEO QUALITY

Since a video is essentially composed of a sequence of image frames arranged in chronological order, video quality can be subdivided into two aspects: temporal quality and spatial quality. Temporal quality focuses on the consistency and continuity between consecutive video frames, while spatial quality emphasizes aspects such as aesthetic value, image sharpness and the naturalness of the content.

**Subject Consistency (SC).** For the subjects in a video, we assess whether their appearance remains consistent throughout the sequence by computing the cross-frame similarity of DINO (Caron et al., 2021) feature, which serves to evaluate different models' capability in maintaining subject consistency.

**Background Consistency (BC).** For the video's overall background, we evaluate the temporal consistency of the background scene by computing the cross-frame similarity of the CLIP (Radford et al., 2021) feature.

**Temporal Flickering (TF).** We observe that videos produced by many editing models exhibit temporal flickering. Accordingly, we quantify temporal flicker by sampling frames and computing the mean absolute difference across frames.

**Motion Smoothness (MS).** Motion smoothness is utilized to evaluate the continuity and naturalness of subject or camera movements. Under normal circumstances, a video should be free from jitter and unnatural acceleration variations. We adopt the motion priors from the video frame interpolation model (Li et al., 2023) to assess the smoothness of motion in the edited videos.

**Video Training Suitability Score (VTSS)** (Wang et al., 2025b) is the output of a supervised model trained on human-annotated data. It integrates indicators such as compositional coherence, aesthetic quality, image sharpness, color saturation, content naturalness, and motion stability, thereby enabling a comprehensive assessment of a video's spatial quality.

## 4.2 INSTRUCTION COMPLIANCE

Instruction compliance is used to evaluate whether the generated target video correctly fulfills the requirements specified in the editing prompt, and whether it is semantically aligned with the target prompt. In addition to general metrics and task-specific criteria for different editing tasks, we further employ MLLM to assist in assessing the semantic consistency between the video content and the editing instructions, thereby enhancing the comprehensiveness and objectivity of the evaluation.

**Overall Semantic Consistency (OSC).** Global semantic consistency is used to holistically evaluate the semantic correspondence between the target video's content and the instruction's intent, with an emphasis on the overall scene. Therefore, we employ VideoCLIP-XL2 (Wang et al., 2024) to compute the semantic similarity between the target video and the target prompt.

**Phrase Semantic Consistency (PSC).** Phrase-level editing adherence is used to assess whether the specific phrases or operations in the instruction are accurately reflected in the target video, with greater emphasis on the edited subject. Accordingly, we employ VideoCLIP-XL2 (Wang et al., 2024) to compute the semantic similarity between the target video and the target phrase.

**Instruction Satisfaction (IS).** Since tasks such as subject motion editing, camera motion editing and camera angle editing are difficult to evaluate accurately using traditional methods, we employ Qwen2.5-VL (Bai et al., 2025b) to assist in determining whether the target video has faithfully executed the edit prompt. Specifically, we input both the edit prompt and the target video into the model, instructing it to assign a score on a five-point scale to indicate the accuracy of execution. Furthermore, we provide detailed descriptions for each score level to ensure that the model maintains consistent evaluation criteria across multiple rounds of assessment.

**Quantity Accuracy (QA).** Quantity correctness is a metric specifically designed for quality editing tasks. This metric uses the target span as input to Grounding DINO (Liu et al., 2024b), compares the number of detected bounding boxes with the quantity specified in the edit prompt, and assigns a score of 1 for correctness and 0 for incorrectness.

Table 1: **Attributes comparison with open-source video editing benchmarks.** Our proposed IVEBench boasts distinct advantages across various key dimensions.

| Method | Video Collection | | Prompt Type | | | | Evaluation Metrics | | | Year |
|---|---|---|---|---|---|---|---|---|---|---|
| | Video Count | Prompt Count | Quantity Editing | Subject Motion Editing | Camera Editing (Motion and Angle) | Visual Effect Editing | Instruction Compliance | Video Fidelity | MLLM | |
| VE-Bench | 169 | 148 | ✗ | ✗ | ✗ | ✗ | ✔ | ✔ | ✗ | 2025 |
| EditBoard | 40 | 80 | ✗ | ✗ | ✗ | ✗ | ✔ | ✔ | ✗ | 2025 |
| VACE-Benchmark | 240 | 480 | ✗ | ✔ | ✗ | ✗ | ✔ | ✔ | ✗ | 2025 |
| FiVE | 100 | 420 | ✗ | ✗ | ✗ | ✗ | ✔ | ✔ | ✔ | 2025 |
| TDVE-Assessor | 180 | 340 | ✗ | ✔ | ✗ | ✗ | ✔ | ✔ | ✔ | 2025 |
| **IVEBench** | **600** | **600** | ✔ | ✔ | ✔ | ✔ | ✔ | ✔ | ✔ | 2025 |

## 4.3 VIDEO FIDELITY

Fidelity is utilized to assess whether the target video retains the unedited portions of the source video, thereby ensuring that the editing process does not introduce irrelevant alterations. In addition to devising conventional metrics from the perspectives of motion and semantics, we further leverage MLLM to assess the content fidelity of the target video, enhancing the robustness of the metric on more challenging tasks.

**Semantic Fidelity (SF).** To quantify the degree of semantic preservation in the target video, we employ VideoCLIP-XL2 (Wang et al., 2024) to compute the feature similarity between the source and target videos.

**Motion Fidelity (MF).** Existing video motion detection often relies on optical flow, but it struggles with occlusions. Therefore, we employ Cotracker3 (Karaev et al., 2024), which is capable of handling occlusions, for extracting reliable motion trajectories. The details of the trajectory similarity computation are provided in Appendix B.

**Content Fidelity (CF).** For tasks such as camera movement editing, camera angle editing and transition editing, the same subject may display different orientations due to variations in perspective, which makes it difficult for traditional metrics to adequately capture content preservation. To address this limitation, we use Qwen2.5-VL (Bai et al., 2025b) to assist in evaluating whether the target video correctly retains those elements that should remain unedited. Specifically, we input the source prompt, the edit prompt, and the target video into the model, instructing it to assign a score on a five-point scale reflecting the fidelity of the unedited content. In addition, we provide detailed descriptions for each score level to ensure that the model adheres to consistent evaluation standards across multiple rounds of assessment.

Among these 12 metrics, SC, BC, TF, and MS are adopted from VBench (Huang et al., 2024). We conducted independence tests on the remaining 8 metrics (detailed in Appendix G), demonstrating that the IVEBench metrics exhibit a high degree of independence.

## 4.4 HUMAN ALIGNMENT FOR BENCHMARK VALIDATION

We select three video editing models $\{A, B, C\}$ and provide 30 source videos with corresponding editing instructions. For a given source video $v_i$ and its editing instruction $p_i$, each selected video editing model produces an edited video, resulting in a set $G_i = \{V_{i,A}, V_{i,B}, V_{i,C}\}$. Within each set, the generated videos are compared in pairs, yielding $C_3^2 = 3$ pairwise comparisons. For each evaluation dimension, we prepare detailed guidelines and illustrative examples, and participants receive prior training to ensure a clear understanding of the dimension definitions. In every pairwise comparison, human annotators are instructed to evaluate the videos exclusively with respect to the specified metric (see Fig. 4), and to subjectively judge which video performs better in that dimension, or to mark the pair as "hard to distinguish." We recruit 30 participants to conduct the human annotation. The conclusions of this experiment will be presented in Sec. 6.2, with further details provided in Appendix F.

## 4.5 UNIFIED SCORING FOR BENCHMARK ASSESSMENT

Each evaluation dimension comprises multiple metrics. To assess their relative importance, trained annotators rate the contribution of each metric and dimension. The average ratings are rounded to the nearest integer and used as weights in the scoring formulas. Detailed formulas for dimensions and total score are provided in Appendix C.

Table 2: **Performance comparison of different video editing methods on our benchmark.** Higher values indicate better performance. [†] denotes that certain high-frame videos fail during inference due to out-of-memory issues. [‡] denotes that the method has a fixed maximum frame number, which is lower than the maximum length of the source videos.

| Database | Method | Dimension Performance | | | | Metric Performance | | | | | | | | | | | |
|---|---|---|---|---|---|---|---|---|---|---|---|---|---|---|---|---|---|
| | | Total Score | Video Quality | Instruction Compliance | Video Fidelity | SC | BC | TF | MS | VTSS | OSC | PSC | IS | QA | SF | MF | CF |
| Short | InsV2V | 0.67 | 0.80 | 0.39 | 0.82 | 0.94 | 0.96 | 0.97 | 0.97 | 0.045 | 0.24 | 0.23 | 3.10 | 0.30 | 0.95 | 0.86 | **4.05** |
| | AnyV2V | 0.58 | 0.73 | 0.42 | 0.59 | 0.89 | 0.94 | 0.97 | 0.97 | 0.026 | 0.22 | **0.24** | 3.33 | 0.30 | 0.80 | 0.82 | 2.75 |
| | StableV2V | 0.51 | 0.69 | 0.43 | 0.41 | 0.85 | 0.92 | 0.96 | 0.96 | 0.019 | 0.20 | **0.24** | 3.56 | 0.20 | 0.75 | 0.75 | 1.79 |
| | VACE[‡] | 0.63 | 0.80 | 0.25 | **0.83** | 0.95 | **0.98** | 0.98 | 0.98 | 0.045 | 0.23 | 0.22 | 2.16 | 0.20 | **0.97** | **0.89** | 4.03 |
| | Lucy-Edit-Dev | 0.64 | **0.82** | 0.34 | 0.75 | 0.95 | 0.96 | 0.98 | 0.99 | **0.051** | 0.24 | 0.22 | 2.84 | 0.20 | 0.93 | 0.68 | 3.83 |
| | Omni-Video[‡] | 0.59 | 0.78 | 0.44 | 0.54 | **0.96** | 0.97 | 0.98 | 0.99 | 0.038 | 0.22 | 0.23 | 3.36 | **0.40** | 0.81 | 0.51 | 2.85 |
| | ICVE[‡] | 0.60 | 0.71 | 0.45 | 0.64 | 0.95 | 0.97 | **0.99** | **1.00** | 0.017 | 0.23 | 0.23 | 3.62 | 0.30 | 0.85 | 0.46 | 3.55 |
| | Ditto[‡] | **0.67** | 0.78 | **0.49** | 0.73 | **0.96** | **0.98** | 0.97 | 0.99 | 0.038 | **0.25** | **0.24** | **3.87** | 0.30 | 0.89 | 0.79 | 3.64 |
| Long | InsV2V | 0.66 | 0.80 | 0.37 | 0.79 | 0.90 | 0.94 | 0.98 | 0.98 | 0.048 | **0.24** | 0.23 | 3.10 | 0.20 | 0.95 | 0.68 | **4.13** |
| | AnyV2V[†] | 0.55 | 0.72 | 0.36 | 0.57 | 0.84 | 0.92 | 0.97 | 0.97 | 0.029 | 0.22 | 0.23 | 3.25 | 0.00 | 0.80 | 0.82 | 2.65 |
| | StableV2VE[†] | 0.51 | 0.69 | 0.42 | 0.41 | 0.83 | 0.91 | 0.96 | 0.96 | 0.021 | 0.23 | 0.23 | 3.45 | **0.25** | 0.70 | 0.77 | 1.79 |
| | VACE[‡] | 0.62 | 0.80 | 0.27 | 0.78 | 0.92 | 0.95 | 0.96 | 0.96 | 0.048 | 0.24 | 0.22 | 2.27 | 0.20 | 0.96 | **0.96** | 3.74 |
| | Lucy-Edit-Dev | 0.65 | **0.82** | 0.32 | **0.81** | 0.91 | 0.95 | 0.98 | 0.99 | **0.053** | 0.24 | 0.22 | 2.65 | 0.20 | **0.97** | 0.73 | 4.13 |
| | Omni-Video[‡] | 0.57 | 0.78 | 0.42 | 0.51 | 0.94 | 0.96 | 0.97 | 0.98 | 0.039 | 0.22 | 0.23 | 3.53 | 0.20 | 0.81 | 0.55 | 2.59 |
| | ICVE[‡] | 0.59 | 0.72 | 0.40 | 0.64 | 0.95 | 0.97 | **0.99** | **1.00** | 0.019 | 0.23 | 0.23 | 3.63 | 0.00 | 0.86 | 0.48 | 3.48 |
| | Ditto[‡] | **0.66** | 0.78 | **0.48** | 0.72 | **0.96** | **0.98** | 0.97 | 0.99 | 0.038 | 0.23 | **0.24** | **3.93** | 0.20 | 0.86 | 0.73 | 3.69 |

## 5 DISCUSSION WITH RECENT VIDEO EDITING BENCHMARKS

Existing video editing benchmarks are primarily designed for source-target prompt-based methods, and they either fail to support or can only minimally accommodate IVE methods (Li et al., 2025b). As summarized in Tab. 1, these benchmarks exhibit clear limitations in dataset scale and coverage. More critically, their prompt design largely remains confined to image editing types (subject editing, attribute editing, or style editing) without dedicated task formulations that address the temporal nature of video. In contrast, IVEBench provides a comprehensive and instruction-centered evaluation suite that introduces three substantial advances: *1)* A large-scale dataset of 600 videos, covering 35 topics across 7 dimensions, with lengths ranging from 32 to 1024 frames, organized into short and long subsets to enhance source diversity and semantic coverage. *2)* Full coverage of eight major categories and thirty-five subcategories of editing tasks, including those that explicitly leverage the unique properties of video, spanning different levels of granularity as well as tasks involving both single and multiple subjects. *3)* MLLM-based metrics specifically designed for Instruction Compliance and Video Fidelity, coupled with human-annotated weightings and dimensions scoring formulas. These innovations enable IVEBench to surpass existing benchmarks in video collection, task coverage, and evaluation methodology, thereby establishing a systematic and practically relevant standard for IVE.

## 6 BENCHMARKING VIDEO EDITING METHOD IN IVEBENCH

### 6.1 EXPERIMENTAL SETUP

We evaluate state-of-the-art IVE models InsV2V (Cheng et al., 2023), AnyV2V (Ku et al., 2024) and StableV2V (Liu et al., 2024a), Lucy-Edit-Dev (Team, 2025), Omni-Video (Tan et al., 2025), ICVE (Liao et al., 2025), Ditto (Bai et al., 2025a) as well as the multi-conditional video editing model VACE (Jiang et al., 2025) using IVEBench, all employed with their official implementations and pretrained weights. Evaluations are conducted on the IVEBench Database. Model-specific configurations, hardware requirements, treatment of failure cases, and evaluation details are described in Appendix D.

### 6.2 BENCHMARKING STATE-OF-THE-ART METHODS ON IVEBENCH

**Quantitative analysis.** From the numerical results in Tab. 2 and Tab. 3 as well as the visualizations in Fig. 4, it can be observed that eight evaluated methods demonstrate relatively good frame-to-frame consistency. However, the per-frame image quality remains unsatisfactory, which consequently leads to low Video Fidelity scores. Moreover, these methods achieve very limited performance in instruction adherence, primarily due to the narrow range of task types they support. Among

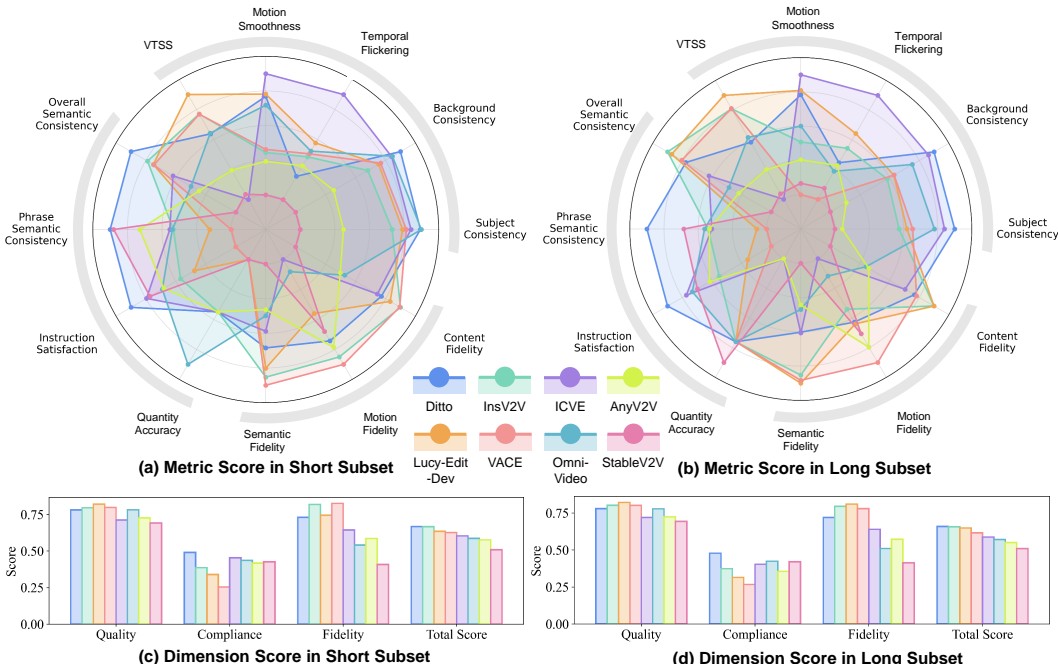

Figure 4: **IVEBench Evaluation Results of Video Editing Models.** We visualize the evaluation results of eight IVE models in 12 IVEBench metrics. We normalize the results per dimension for clearer comparisons. For comprehensive numerical results, please refer to Tab. 2.

them, Lucy-Edit-Dev (Team, 2025) exhibits the best performance in editing speed. Ditto (Bai et al., 2025a) and InsV2V (Cheng et al., 2023) demonstrate the best performance in terms of editing capability, as they can execute editing prompts while preserving visual integrity. Furthermore, since Ditto successfully handles a larger number of cases, it achieves the best performance in Instruction Compliance. Nevertheless, these models achieve a Total Score of no more than 0.7 and an Instruction Compliance score of no more than 0.5, indicating that existing IVE methods still have substantial room for improvement in overall editing capability, particularly in Instruction Compliance.

**Qualitative analysis.** As illustrated in Fig. 5, the outputs of different models reveal consistent weaknesses across multiple editing scenarios. First, all models tend to introduce inaccurate localization of the desired edit, leading to visible artifacts such as geometric distortion, semantic bleeding, semantic collapsing, boundary blurring, and texture flickering. These artifacts significantly compromise the per-frame visual quality of edited videos, which in turn diminishes their overall fidelity. Second, when observing more challenging editing types, such as subject motion editing and camera angle editing, we find that the editing capability of current models is particularly limited, underscoring an urgent need for broader task coverage in future development. Moreover, the models show distinct

Table 3: Inference efficiency and resolution.

| Database | Method | Time per Frame↓ | Max Memory↓ | Video Resolution |
|---|---|---|---|---|
| **Short** | InsV2V | 3.96s | **12.81GB** | 512×512 |
| | AnyV2V | 11.66s | 27.37GB | 512×512 |
| | StableV2V | 3.90s | 28.31GB | 512×512 |
| | VACE[‡] | 27.03s | 122.18GB | **1280×720** |
| | Lucy-Edit-Dev | **1.52s** | 32.21GB | 832×480 |
| | Omni-Video[‡] | 1.8os | 36.37GB | 640×352 |
| | ICVE[‡] | 5.05s | 88.33GB | 384×240 |
| | Ditto[‡] | 19.69s | 38.49GB | 832×480 |
| **Long** | InsV2V | 4.05s | **13.48GB** | 512×512 |
| | AnyV2V[†] | 11.47s | 63.15GB | 512×512 |
| | StableV2V[†] | 3.72s | 49.82GB | 512×512 |
| | VACE[‡] | 51.00s | 132.90GB | **1280×720** |
| | Lucy-Edit-Dev | 2.23s | 34.33GB | 832×480 |
| | Omni-Video[‡] | **2.04s** | 36.37GB | 640×352 |
| | ICVE[‡] | 5.28s | 91.67GB | 384×240 |
| | Ditto[‡] | 20.15s | 38.86GB | 832×480 |

behavioral patterns: StableV2V (Liu et al., 2024a) often applies overly aggressive modifications that satisfy the editing prompt but neglect the preservation of unedited content; InsV2V (Cheng et al., 2023), in contrast, tends to adopt a conservative strategy, retaining much of the source content when dealing with unfamiliar instructions; while VACE (Jiang et al., 2025), not being a native IVE model, frequently fails to properly execute the given edits, resulting in weak compliance with the prompts. These qualitative findings highlight that improving per-frame image fidelity and expanding editing

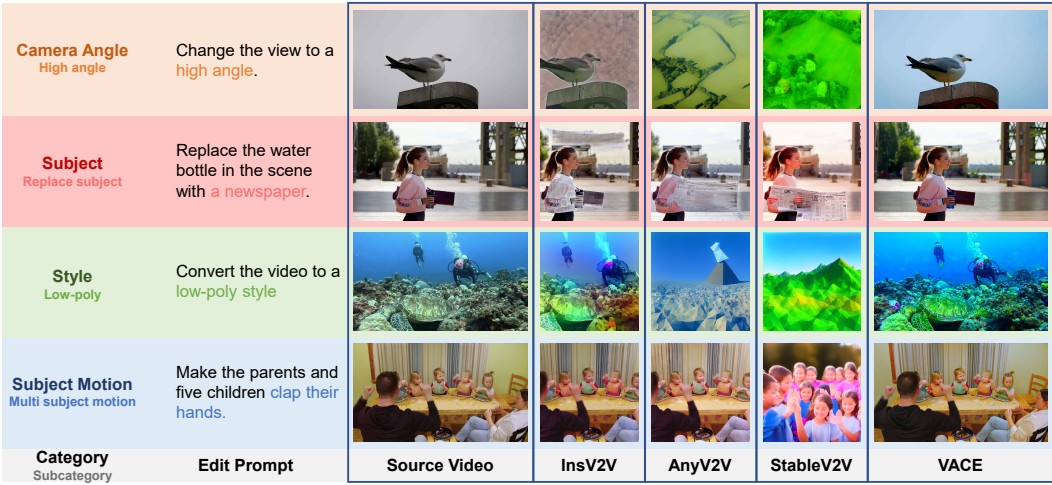

Figure 5: Qualitative comparison of selected IVE methods.

versatility are essential directions for advancing IVE models. More detailed qualitative comparisons and analyses can be found in Appendix E.

Table 4: **Spearman's Rho ($\rho$) across different metrics.** These scores show that IVEBench metrics are highly aligned with human judgments.

| | Video Quality | | | | | Instruction Compliance | | | | Video Fidelity | | |
|---|---|---|---|---|---|---|---|---|---|---|---|---|
| | SC | BC | TF | MS | VTSS | OSC | PSC | IS | QA | SF | MF | CF |
| $\rho$ | 0.9536 | 0.9503 | 0.8842 | 0.9774 | 0.9985 | 0.7210 | 0.8316 | 0.9834 | 0.8104 | 0.9453 | 0.9371 | 0.9892 |

**Human alignment results.** To validate that our evaluation metrics align with human perception, as described in Sec. 4.4, we conduct human annotations for each metric. In pairwise model comparisons, the preferred model is assigned a score of 1, while the other receives 0. If annotators express no preference, both models are assigned 0.5. For each metric, a model's final human score is computed as the total score divided by the number of comparisons. We then calculate Spearman's rank correlation coefficient between these human scores and the automatic evaluation metric scores. The results in Tab. 4 demonstrate that our proposed evaluation metrics exhibit a high degree of consistency with human preferences. Furthermore, to ensure the inter-rater reliability of the evaluation, we calculated Fleiss' Kappa for our human annotations and obtained a score of 0.78, which indicates 'substantial agreement' according to Landis and Koch (Landis & Koch, 1977). This demonstrates that our rigorous calibration process effectively aligned the standards of different annotators.

## 6.3 INSIGHTS AND DISCUSSIONS

**High frame-to-frame consistency, weak single-frame quality.** Across models, frame-to-frame consistency is generally well preserved, with limited temporal flickering. However, the quality of individual frames often shows frequent visible artifacts such as semantic bleeding, boundary blurring and texture flickering. These issues also lead to a noticeable degradation in Video Fidelity, highlighting the necessity for future work to develop effective strategies to mitigate such artifacts.

**Limited support for diverse editing prompt types.** Models perform poorly in the Compliance dimension mainly because they only handle a few basic editing types reasonably well, namely subject editing, style editing and attribution editing. In contrast, they lack the capacity to execute more advanced editing types such as quantity editing, subject motion editing, visual effect editing, camera motion editing and camera angle editing. This leads to consistently low scores across all compliance-related metrics. Future video editing models should therefore place greater emphasis on broadening the range of supported editing prompts.

**Intrinsic limitations of first-frame-based editing models.** Video editing, unlike image editing, requires maintaining temporal coherence, which introduces the need to modify middle or later frames

of a video. For example, to handle transitions or to insert intermediate events. These tasks do not originate from modifications in the initial frames but instead focus on transformations that occur later in the sequence. First-frame-based models, however, propagate changes from the beginning throughout the entire video, making them inadequate for such editing requirements.

**Scalability to long video sequences.** A critical challenge in IVE lies in handling long sequences with hundreds or even thousands of frames. Most existing methods, especially those relying on frame-wise diffusion or first-frame propagation, exhibit a near-linear growth in GPU memory consumption and latency as sequence length increases, making them impractical for videos beyond 128 frames. In contrast, InsV2V (Cheng et al., 2023) demonstrates superior scalability by adopting a chunked inference strategy with latent overlap, where only a limited set of reference frames is preserved across segments. This design effectively constrains memory growth while maintaining temporal continuity.

**Resolution limitations.** Existing IVE methods, typically operate at $512 \times 512$ or $832 \times 480$ resolution, which is far below the standard of real-world user content. The multi-conditional video editing model VACE (Jiang et al., 2025) can support 720P outputs; however, this still falls short of the practical demand, as user videos are commonly recorded in 1080P or higher resolutions, and the expectation is that edited outputs should preserve this level of detail. The low-resolution setting limits visual fidelity, which results in artifacts such as blurred textures and edge degradation, and also reduces usability in professional media workflows.

## 7 CONCLUSION

With the rapid progress of IVE, how to systematically and comprehensively evaluate these methods has become a central challenge in the field. Existing benchmarks exhibit clear limitations in terms of video source diversity, task coverage, and evaluation dimensions, making them insufficient to reliably reflect the true capabilities of current approaches or to provide meaningful guidance for subsequent research. To address these issues, we propose IVEBench, a modern benchmarking suite designed for IVE models. IVEBench integrates a large-scale and diverse dataset, a broad range of editing tasks, and a multi-dimensional evaluation protocol that leverages MLLMs and aligns closely with human perception. We expect IVEBench to play a key role in the evaluation of video editing models and in advancing the development of the field.

**Limitation and future work.** As more IVE models are released as open source, we plan to incorporate them into IVEBench for further benchmarking and comparison. In the future, with the increase of computational power, we will also expand the scale of data used for evaluation.

## ETHICS STATEMENT

This work complies with the ICLR Code of Ethics. Our dataset is built exclusively from publicly available, open-licensed video sources, and all annotations were conducted with careful respect for privacy and fairness. No personally identifiable or sensitive information is included. We release both data and code to promote transparency, reproducibility, and responsible stewardship of research in instruction-guided video editing. The dataset and code are released solely for the purpose of advancing academic research.

## REPRODUCIBILITY STATEMENT

We have already elaborated on all the models or algorithms proposed, experimental configurations, and benchmarks used in the experiments in the main body or appendix of this paper. Furthermore, we declare that the entire code used in this work will be released after acceptance.

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

APPENDIX

OVERVIEW

The supplementary material presents more comprehensive results of our IVEBench to facilitate the comparison of subsequent benchmarks:

- **Appendix A** provides more detailed descriptions of edit prompt subcategories, accompanied by concrete examples.
- **Appendix B** provides the detailed procedure for computing motion fidelity score
- **Appendix C** provides the unified scoring formulation and detailed explanations of the weighting strategy across metrics and dimensions
- **Appendix D** provides experimental details, including hardware configurations, dataset partitioning for evaluation, model implementations and failed video IDs.
- **Appendix E** provides a detailed comparison of model performance.
- **Appendix F** provides human alignment details, including annotator guideline design and annotation interface.
- **Appendix G** provides independence analysis for the metrics introduced in IVEBench.
- **Appendix H** provides information on the use of LLMs.

## A   DESCRIPTIONS OF VARIOUS CATEGORIES OF EDIT PROMPTS

In this section, we provide detailed descriptions of all 35 subcategories of editing prompts included in IVEBench. Each subcategory is defined with its specific editing operation and supported by a representative example to illustrate how the editing request is expressed. The purpose of this collection is to ensure clarity, reproducibility, and comprehensive coverage of diverse instruction-guided video editing tasks. Tab. A1 summarizes the categories, subcategories, descriptions, and corresponding examples for ease of reference.

Table A1: **Description and example for each subcategory.** We provide detailed descriptions of 35 subcategories along with corresponding examples to facilitate understanding.

| Category | Subcategory | Description | Example |
|---|---|---|---|
| Style Editing | watercolor | Apply watercolor painting style to video | Convert the video to a watercolor style |
| Style Editing | pixel | Convert video to retro pixel art style | Convert the video to a pixel-style |
| Style Editing | anime | Render video in anime style | Change the style of the video to anime |
| Style Editing | American comic style | Apply American comic book style | Transform the video into a American comic style |
| Style Editing | ukiyo-e | Render video in Japanese ukiyo-e style | Convert the video style to ukiyo-e |
| Style Editing | black and white | Convert video to black-and-white tones | Convert the video to black and white |
| Style Editing | oil painting | Apply oil painting effect to video | Transform the video into an oil painting style |
| Style Editing | cyberpunk | Render video in neon futuristic cyberpunk style | Convert the video to a cyberpunk style |
| Style Editing | Ghibli | Apply Studio Ghibli-inspired animation style | Change the video style to Ghibli style |

| Category | Subcategory | Description | Example |
|---|---|---|---|
| Style Editing | low-poly | Convert video to simplified low-poly visuals | Transform the video into a low-poly style |
| Style Editing | weather shift | Change weather conditions in the video | Change the weather to a torrential downpour |
| Subject Editing | add new subject | Add a new subject into the video | Add a heron standing among the reeds |
| Subject Editing | remove existing subject | Remove a subject from the video | Remove the young child from the video |
| Subject Editing | replace existing subject | Replace one subject with another | Replace the grotesque creatures with friendly fairy-like beings |
| Attribute Editing | color adjustment | Adjust colors of video or subjects | Change the sky to a deep red color |
| Attribute Editing | subject scaling | Resize a subject in the video | Scale up the man dressed in ancient Egyptian attire |
| Attribute Editing | position change | Change subject position in the scene | Move the girl to the left side of the stone steps |
| Subject Motion Editing | single subject motion | Animate or adjust one subject's motion | Make the man in the black leather jacket stand up and stretch |
| Subject Motion Editing | multiple subject motion | Animate or adjust multiple subjects' motions | Make the woman and the man cry and wipe their tears with their hands |
| Camera Motion Editing | dolly in | Simulate camera moving forward | Move the camera closer to the man in the black shirt |
| Camera Motion Editing | dolly out | Simulate camera moving backward | Gradually move the camera away from the group of men |
| Camera Motion Editing | tracking | Simulate camera following a subject | Track the movement of the red powder as it falls into the bottle |
| Camera Motion Editing | boom up | Simulate camera moving upward | Perform a boom up shot on the white Toyota SUV driving up the dirt hill |
| Camera Motion Editing | arc shot | Simulate camera circling around a subject | Perform an arc shot around the tram as it arrives at the station |
| Camera Motion Editing | zoom in | Zoom in on the video subject | Zoom in on the slice of yellow cake being lifted |
| Camera Motion Editing | zoom out | Zoom out to show more scene | Gradually move the camera away to the ancient temple |
| Camera Angle Editing | high angle | View subject from a high angle | Change the view to a high angle |
| Camera Angle Editing | low angle | View subject from a low angle | Change the view to a low angle |
| Camera Angle Editing | front view | Show subject from the front | Change the view to a front view |
| Camera Angle Editing | side view | Show subject from the side | Change the view to a side view |
| Quantity Editing | increase | Increase number of subjects | Increase the number of A woman with a tattoo to 2 |
| Quantity Editing | decrease | Decrease number of subjects | Decrease the number of flowers to 1 |
| Visual Effect Editing | transition | Add transition between video contents | A particle effect transition, the man wearing sunglasses and smiling |
| Visual Effect Editing | decoration effect | Add decorative visual overlays | Add a flame effect to the metal file |

| Category | Subcategory | Description | Example |
|---|---|---|---|
| Visual Effect Editing | event effect | Add event-based effects | The man turns into sand and is blown away |

## B    MOTION FIDELITY COMPUTATION DETAILS

We describe the computation of motion fidelity between a source video and a target video. Given a video sequence, we sample query points on a uniform grid of size $g$. For each query point $p$, CoTracker3 (Karaev et al., 2024) outputs a trajectory $\mathbf{x}^p = (\mathbf{x}_1^p, \mathbf{x}_2^p, \ldots, \mathbf{x}_T^p)$ with $\mathbf{x}_t^p \in \mathbb{R}^2$, together with a visibility vector $\mathbf{v}^p = (v_1^p, v_2^p, \ldots, v_T^p)$ where $v_t^p \in [0, 1]$ indicates whether $p$ is visible at frame $t$. To compare two videos of different lengths, all trajectories are interpolated to a synchronized length $T = \min(T_1, T_2)$ using linear interpolation based on visible frames.

Given two synchronized tracks $(\tilde{\mathbf{x}}^p, \tilde{\mathbf{v}}^p)$ and $(\tilde{\mathbf{y}}^q, \tilde{\mathbf{w}}^q)$, we compute the frame-wise position distance

$$d_t^{\text{pos}} = \|\tilde{\mathbf{x}}_t^p - \tilde{\mathbf{y}}_t^q\|_2,$$

and velocity distance

$$d_t^{\text{vel}} = \|(\tilde{\mathbf{x}}_t^p - \tilde{\mathbf{x}}_{t-1}^p) - (\tilde{\mathbf{y}}_t^q - \tilde{\mathbf{y}}_{t-1}^q)\|_2 \quad \text{for } t > 1,$$

with $d_1^{\text{vel}} = d_2^{\text{vel}}$. Both distances are normalized by the average spatial span of the tracks

$$\alpha = \tfrac{1}{2} \left( \| \max_t \tilde{\mathbf{x}}_t^p - \min_t \tilde{\mathbf{x}}_t^p \|_2 + \| \max_t \tilde{\mathbf{y}}_t^q - \min_t \tilde{\mathbf{y}}_t^q \|_2 \right),$$

with $\alpha \geq 10^{-6}$. We then define normalized distances $\hat{d}_t^{\text{pos}} = d_t^{\text{pos}}/\alpha$, $\hat{d}_t^{\text{vel}} = d_t^{\text{vel}}/\alpha$ and convert them into similarities $s_t^{\text{pos}} = 1/(1 + \hat{d}_t^{\text{pos}})$, $s_t^{\text{vel}} = 1/(1 + \hat{d}_t^{\text{vel}})$. The frame-wise similarity is obtained by weighted combination

$$s_t = 0.7 s_t^{\text{pos}} + 0.3 s_t^{\text{vel}},$$

and further weighted by visibility $w_t = \min(\tilde{v}_t^p, \tilde{w}_t^q)$. The overall track similarity is

$$S(p, q) = \begin{cases} \dfrac{\sum_{t=1}^T s_t w_t}{\sum_{t=1}^T w_t}, & \text{if } \sum_t w_t > 0, \\ 0, & \text{otherwise.} \end{cases}$$

Let $N_1$ and $N_2$ be the numbers of valid tracks in the source and target videos. We construct a similarity matrix $\mathbf{M} \in \mathbb{R}^{N_1 \times N_2}$ with $\mathbf{M}_{ij} = S(p_i, q_j)$. To establish correspondence, we apply the Hungarian algorithm to maximize $\sum_i \mathbf{M}_{i,\pi(i)}$ with one-to-one mapping $\pi$. We discard pairs with $\mathbf{M}_{i,\pi(i)} \leq 0.3$ and compute the final motion fidelity between videos $V_1$ and $V_2$ as

$$\text{MF}(V_1, V_2) = \frac{1}{|\mathcal{P}|} \sum_{i \in \mathcal{P}} \mathbf{M}_{i,\pi(i)},$$

where $\mathcal{P} = \{i \mid \mathbf{M}_{i,\pi(i)} > 0.3\}$ is the set of valid correspondences. Finally, given $K$ video pairs, the dataset-level motion fidelity score is

$$\text{MF} = \frac{1}{K} \sum_{k=1}^K \text{MF}(V_{\text{src}}^{(k)}, V_{\text{tgt}}^{(k)}).$$

Here, $T$ is the number of synchronized frames, $\mathbf{x}_t^p \in \mathbb{R}^2$ is the 2D position of track $p$ at time $t$, $v_t^p$ is its visibility, $d_t^{\text{pos}}$ and $d_t^{\text{vel}}$ are frame-wise distances, $s_t \in [0, 1]$ is the frame-wise similarity, $S(p, q)$ is the similarity of two tracks, $\mathbf{M}_{ij}$ the similarity matrix, and $\pi$ the matching permutation given by the Hungarian algorithm.

## C  UNIFIED SCORING FORMULATION AND DETAILS

For a given evaluation dimension $D$, let the set of metrics be $\{m_1, m_2, \ldots, m_{n_D}\}$ with corresponding weights $\{w_1, w_2, \ldots, w_{n_D}\}$. The score for dimension $D$ is defined as:

$$S_D = \frac{\sum_{i=1}^{n_D} w_i \cdot m_i}{\sum_{i=1}^{n_D} w_i}.$$

In our study, the three dimensions are computed as:

$$\text{Video Quality} = \frac{\sum_{i \in \mathcal{V}} w_i \cdot m_i}{\sum_{i \in \mathcal{V}} w_i},$$

$$\text{Instruction Compliance} = \frac{\sum_{i \in \mathcal{I}} w_i \cdot m_i}{\sum_{i \in \mathcal{I}} w_i},$$

$$\text{Video Fidelity} = \frac{\sum_{i \in \mathcal{F}} w_i \cdot m_i}{\sum_{i \in \mathcal{F}} w_i}.$$

where $\mathcal{V}$, $\mathcal{I}$, and $\mathcal{F}$ denote the sets of metrics belonging to *Video Quality*, *Instruction Compliance*, and *Video Fidelity*, respectively. The overall score is obtained by treating the three dimension scores as higher-level metrics. Let the set of dimensions be $\mathcal{D}$, with scores $\{S_j\}$ and corresponding weights $\{\alpha_j\}$. The total score is given by:

$$\text{Total Score} = \frac{\sum_{j \in \mathcal{D}} \alpha_j \cdot S_j}{\sum_{j \in \mathcal{D}} \alpha_j}.$$

Both metric-level weights $w_i$ and dimension-level weights $\alpha_j$ are determined from the user study: each participant provided importance ratings (0-5) for metrics and dimensions. The ratings were averaged across participants, rounded to the nearest integer, and applied directly in the above formulas. According to the participants' ratings, the weight of VTSS is 5, the weights of IS and CF are 3, while the weights of other metrics are 1. Since the weights of the three dimensions are all 4, they are normalized to 1. The resulting scores for different models are reported in Tab. 2.

## D  EXPERIMENT DETAILS

All experiments were performed on NVIDIA H20 GPUs: InsV2V, AnyV2V, and StableV2V were run on a single GPU, while VACE required two GPUs for 720P inputs. Each model's output video resolution followed its officially recommended setting, and the number of generated frames was matched to the input sequence. When certain videos could not be processed due to out-of-memory errors, the corresponding results were excluded, and the indices of these failed videos are provided in Tab. A2. Moreover, due to VACE's fixed maximum frame limit of 81 frames (Jiang et al., 2025), source videos exceeding this length were uniformly sampled to 81 frames specifically for VACE editing. The maximum frame count for ICVE (Liao et al., 2025) is set to the officially recommended 81 frames, as exceeding this limit results in excessive GPU memory usage. Similarly, Omni-Video (Tan et al., 2025) is configured with a maximum of 17 frames to prevent output noise, while Ditto (Bai et al., 2025a) is set to the officially recommended 73 frames. Both short and long subsets of the IVEBench Database were used to assess editing capability across video lengths. For each model and subset, we also recorded the average runtime per frame and the peak GPU memory consumption. Evaluation was carried out using the twelve indicators of IVEBench Metrics, organized into three dimensions, where indicator scores were first computed per task, then averaged across videos, with irrelevant indicators omitted depending on the editing type; finally, all scores were normalized before visualization.

Table A2: **Failed video count and IDs of IVE methods.** We present the methods that cause GPU memory usage to exceed the capacity of a single H20 card due to excessively long frame sequences in certain videos.

| Method | Failed videos count | Failed video IDs |
|--------|---------------------|------------------|
| AnyV2V | 65 | long_0001, long_0004, long_0005, long_0007, long_0008, long_0009, long_0010, long_0011, long_0013, long_0014, long_0015, long_0016, long_0018, long_0020, long_0021, long_0022, long_0023, long_0025, long_0028, long_0029, long_0030, long_0031, long_0032, long_0033, long_0034, long_0035, long_0037, long_0039, long_0042, long_0043, long_0053, long_0055, long_0058, long_0059, long_0060, long_0061, long_0064, long_0068, long_0070, long_0074, long_0075, long_0082, long_0088, long_0091, long_0094, long_0095, long_0096, long_0098, long_0100, long_0104, long_0113, long_0114, long_0115, long_0117, long_0125, long_0150, long_0152, long_0153, long_0155, long_0159, long_0169, long_0179, long_0180, long_0186, long_0200 |
| StableV2V | 102 | long_0001, long_0004, long_0005, long_0007, long_0008, long_0009, long_0010, long_0011, long_0012, long_0013, long_0014, long_0015, long_0016, long_0018, long_0020, long_0021, long_0022, long_0023, long_0025, long_0028, long_0029, long_0030, long_0031, long_0032, long_0033, long_0034, long_0035, long_0036, long_0037, long_0038, long_0039, long_0040, long_0041, long_0042, long_0043, long_0044, long_0045, long_0047, long_0049, long_0053, long_0055, long_0057, long_0058, long_0059, long_0060, long_0061, long_0064, long_0067, long_0068, long_0070, long_0071, long_0073, long_0074, long_0075, long_0077, long_0079, long_0082, long_0083, long_0088, long_0089, long_0091, long_0094, long_0095, long_0096, long_0097, long_0098, long_0100, long_0102, long_0104, long_0106, long_0108, long_0110, long_0111, long_0112, long_0113, long_0114, long_0115, long_0117, long_0123, long_0124, long_0125, long_0128, long_0130, long_0150, long_0152, long_0155, long_0159, long_0160, long_0164, long_0168, long_0169, long_0171, long_0173, long_0177, long_0179, long_0180, long_0186, long_0188, long_0195, long_0197, long_0198, long_0200 |

## E  DETAILED QUANTITATIVE COMPARISON AND ANALYSIS

In this section, we provide a more detailed quantitative comparison of the evaluated models across different categories of instruction-guided video editing. While the main results are summarized in Table 2 and Figure 4 of the main paper, here we extend the analysis to highlight model behaviors under specific editing tasks and frame lengths. We further complement the numerical results with Fig. A1, which illustrates representative editing scenarios by concatenating the first, middle, and last

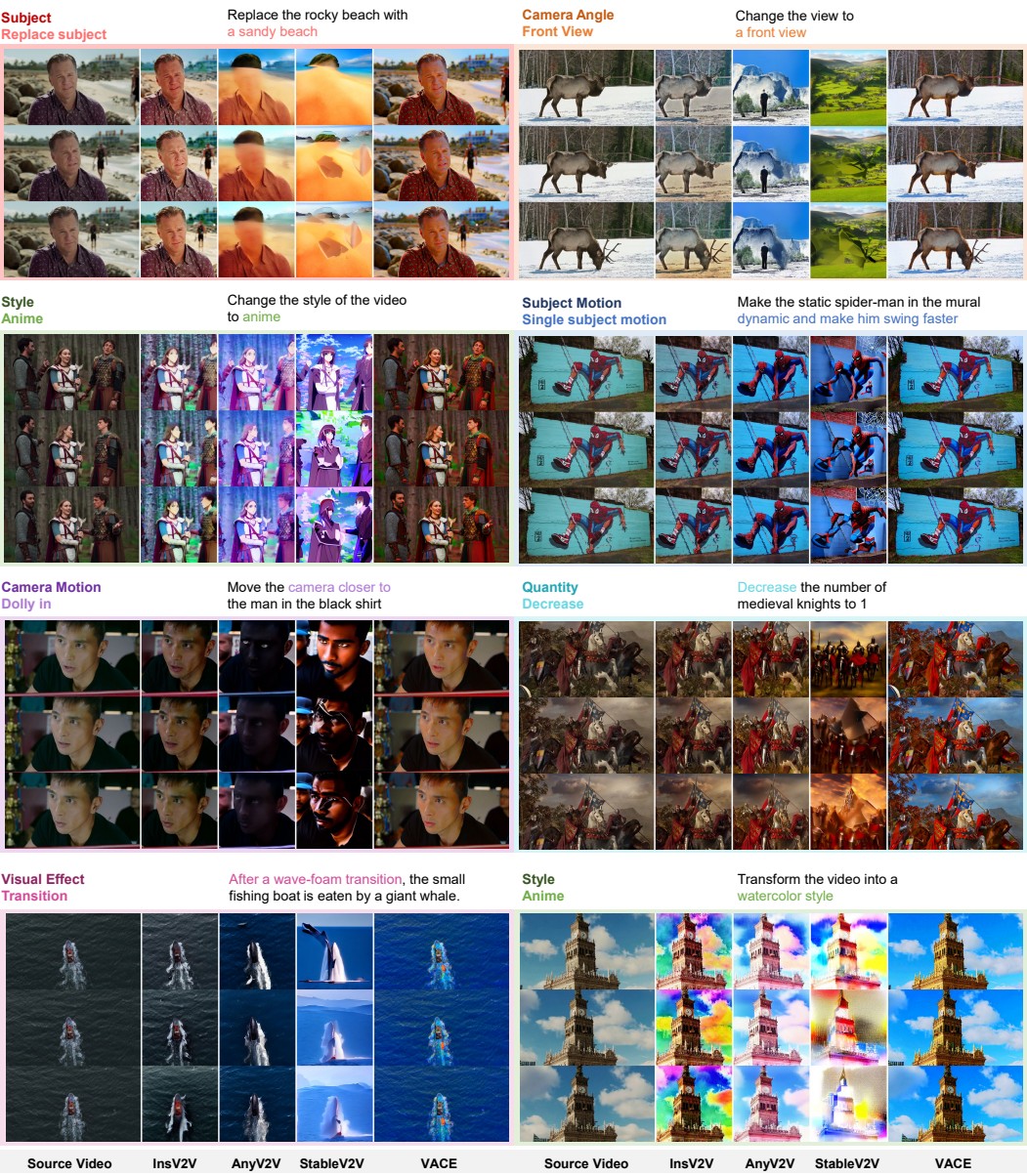

Figure A1: **Visualization of Model Output Comparison.** We concatenate the first, middle, and last frames of the video to facilitate comparison of the temporal performance across different models.

frames of each generated video. This visualization helps reveal temporal dynamics and qualitative differences that may not always be fully captured by scalar metrics.

Specifically, InsV2V demonstrates relatively balanced performance across most categories, maintaining higher semantic fidelity and motion fidelity even in longer sequences. However, its conservative strategy sometimes leads to under-editing, resulting in lower scores in instruction satisfaction. AnyV2V exhibits strong Instruction Compliance in simpler style and attribute editing tasks, yet struggles under difficult editing tasks. The aggressive editing strategy of stableV2V leads to a higher instruction satisfaction score, but visual inspections clearly show severe semantic bleeding and boundary artifacts when dealing with complex prompts. Finally, VACE, though not originally designed for IVE, achieves reasonable temporal smoothness and high resolution outputs; nevertheless, its restricted maximum frame length limits its applicability, and its overall performance in instruction compliance remains unsatisfactory compared to native IVE models.

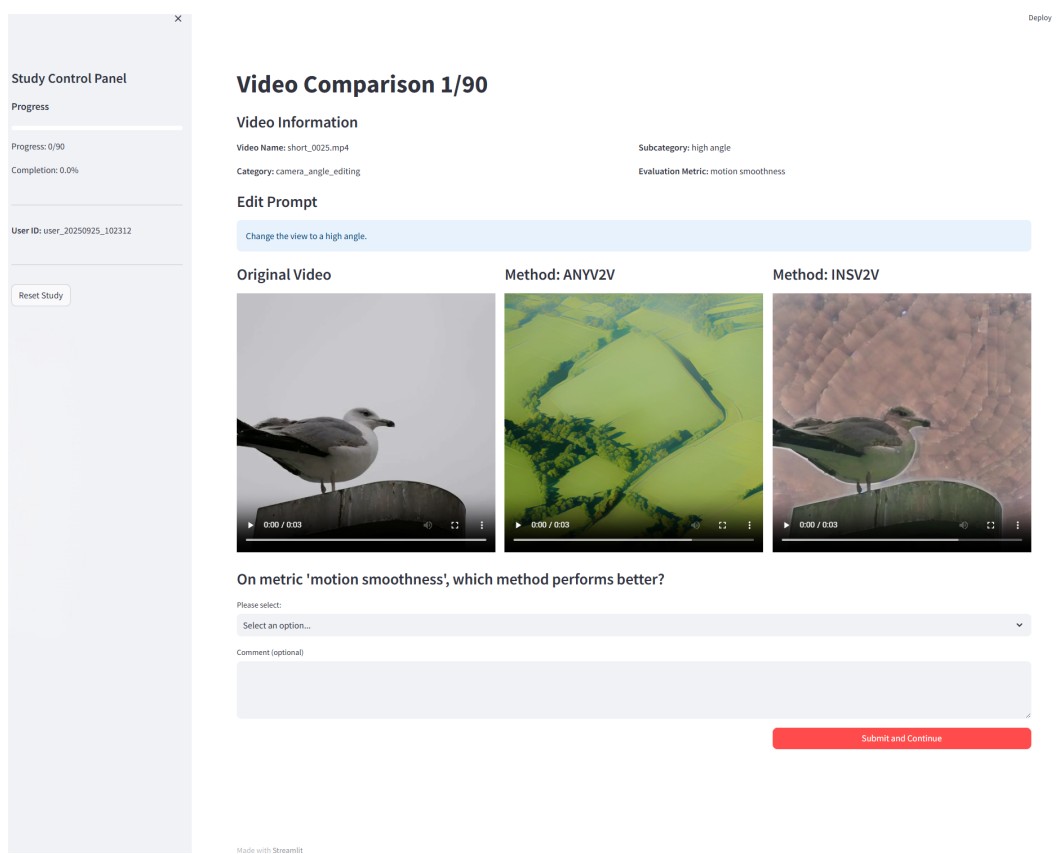

Figure A2: **Human Annotation Interface for Benchmark Validation.** The interface presents the source video, the editing instruction, and the outputs of different models under a specified evaluation dimension, enabling annotators to conduct pairwise comparisons and judge which video better satisfies the given criterion.

Taken together, these detailed results and the examples in Fig. A1 confirm that current models, while capable of maintaining frame-to-frame coherence, still fall short in faithfully executing diverse instructions and preserving high per-frame fidelity. This underscores the necessity of IVEBench in identifying fine-grained weaknesses and providing clear guidance for future methodological improvements.

## F    HUMAN ALIGNMENT DETAILS

To validate the alignment of IVEBench metrics, we first provided each annotator with a detailed explanation of the meaning of each metric along with illustrative examples of good and poor cases, followed by additional case-based tests to ensure that the annotators fully understood the intended interpretation of the metric. Moreover, we conducted further tests to confirm that the annotators focused exclusively on the designated metric during comparisons, rather than being influenced by the overall quality of the videos. For ease of experimentation, we designed a dedicated annotation interface for human evaluators. The interface displays the source video, editing instruction, and the outputs of different models for direct comparison under a specified evaluation dimension. Annotators are instructed to make pairwise comparisons between outputs, choosing the video that better satisfies the designated metric or marking them as indistinguishable when necessary. The design of the interface is illustrated in Fig. A2.

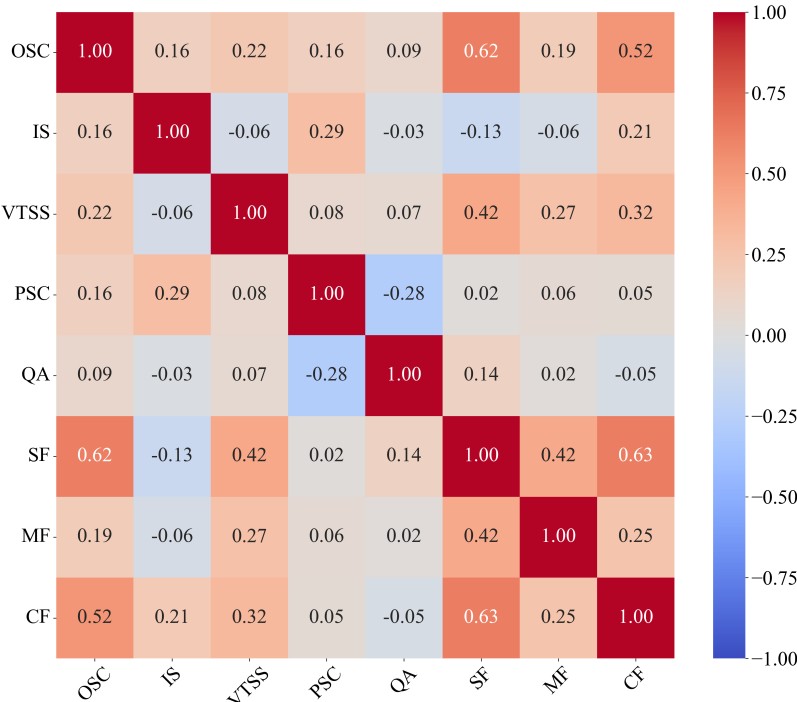

Figure A3: **Heatmap of metric independence coefficients.** We calculated the Spearman correlation coefficient matrix across the results of all the eight IVE models.

## G  INDEPENDENCE OF METRICS

Metric independence is crucial for a benchmark, as it helps avoid redundancy and better reveals the trade-offs among the evaluated models. To quantitatively verify the independence of the 8 metrics proposed in IVEBench (distinct from SC, BC, TF, and MS, which are adopted from VBench (Huang et al., 2024)), we calculated the Spearman correlation coefficient matrix across the results of all models, as shown in Fig. A3. The results demonstrate that the Spearman correlation coefficients among all metrics do not exceed 0.65. This indicates the absence of collinearity (Dormann et al., 2013), supporting the independence and distinctiveness of our metric design. Additionally, negative correlations observed among several metrics suggest that IVEBench metrics can effectively reveal the trade-offs within the evaluated models.

## H  THE USE OF LARGE LANGUAGE MODELS

We use large language models solely for polishing our writing, and we have conducted a careful check, taking full responsibility for all content in this work.

