# OpenReview forum: "IVEBench: Modern Benchmark Suite for Instruction-Guided Video Editing Assessment"
_ICLR.cc/2026/Conference — ICLR 2026 Poster_

### Official Review · Reviewer_xwi2 · 2025-10-25

**Soundness:** 4
**Presentation:** 3
**Contribution:** 3
**Rating:** 4
**Confidence:** 3

**Summary:**

This paper introduces IVEBench, a comprehensive benchmark suite designed to address the shortcomings of existing evaluation methods for instruction-guided video editing. The authors identify three limitations in current benchmarks: insufficient source video diversity, limited scope of editing tasks, and incomplete evaluation metrics. To overcome these challenges, this paper proposed to source more diverse videos, enriching the prompt types and introduce better metrics for evaluation, resulting in the IVEBench benchmark which contains 600 videos.

**Strengths:**

Main contributions:

1.  A new dataset of 600 high-quality, high-resolution source videos, categorized across seven semantic dimensions and split into short (32-128 frames) and long (129-1,024 frames) subsets to test model scalability. The IVEBench includes a set of 600 editing instructions spanning eight major task categories and 35 subcategories, which go beyond simple style/subject edits to include complex temporal tasks like camera motion and subject motion editing.
2.  This paper proposed a three-dimensional evaluation framework assessing **Video Quality**, **Instruction Compliance**, and **Video Fidelity**. This eval framework integrates traditional metrics with MLLMs-based assessments for a more fine-grained and human-aligned evaluation. The Spearman's Rho correlations (e.g., 0.98 for IS, 0.99 for VTSS, 0.94 for SF) provide strong evidence that the metrics are aligned with human perception.

**Weaknesses:**

1. First, the benchmark's construction creates an idealized environment that may not reflect the challenges of real-world video editing. The dataset is composed of high-quality, professional video, which does not represent the noisy in-the-wild videos that users commonly edit. Similarly, the editing prompts are not complex enough (too short and too "clean"), failing to test a model's ability to handle the messy prompts/ This gap between the benchmark's sanitized conditions and real-world complexity limits the generalizability of the performance scores, as models that excel on IVEBench may still falter on more realistic user-generated content and commands.

2. Second, the evaluation protocol, while interesting, has weaknesses related to its reproducibility and objectivity due to its heavy reliance on specific MLLMs - Qwen2.5-VL.  The use of MLLMs in eval, while has some merits as I mentioned, still fall short in lack of explanability, creating a risk of hidden biases. I don't see the novelties of evaluating such a VE task with MLLMs.

3. The benchmark's scope does not fully encompass the complexity and scale of advanced video editing tasks. Especially in terms of the quantity - 600 videos with 35 topics are not enough to cover the complex and dynamic nature of real-world videos. The authors need to clearly explain why only source 600 videos like the cost of creating such benchmarks.

**Questions:**

N/A

**Details Of Ethics Concerns:**

The videos in the benchmark must have appropriate license to be used in research - they are sourced from places like pexels.

---

> ### Author Response · Authors · 2025-11-22
>
> #### Q1: Complex Instructions
>
> Thank you for your insightful review.
>
> - We greatly appreciate your suggestion. Currently, IVEBench includes both simple editing instructions and composite instructions with multiple requirements. For example, the `edit_prompt` for `short_0369` is *"Add a particle effect transition, then show the man wearing sunglasses and smiling."*
> - The current version of the IVEBench dataset contains a significant number of simple instructions, reflecting the early developmental stage of video editing capabilities. Despite the simplicity of the instructions, the latest open-source IVE methods still perform poorly on our metrics (specifically, evaluated open-source models score no higher than **0.5** on the **Instruction Compliance** dimension). Prioritizing complex instructions at this stage would result in excessive difficulty, preventing a clear comparison of editing performance across different models.
> - We plan to expand IVEBench in the future by introducing a dataset focused on composite, multi-step editing instructions to align with the advancement of IVE methods and evaluation standards.
>
>
>
> #### Q2: Video Quality Distribution
>
> Thank you for your insightful review.
>
> - Regarding the content of the video dataset, our sources include **UltraVideo, OpenHumanVid, Pexels, and Mixkit**. UltraVideo data comes from YouTube and contains many videos uploaded by general users (e.g., `short_0061`, `short_0123`, `long_0066`, `long_0085`).
> - Acknowledging the validity of your feedback, we will balance the proportion of professional videos and user-generated content (UGC) in the expanded version of IVEBench to reduce the gap with complex real-world scenarios.
>
>
>
> #### Q3: Potential Bias in MLLMs
>
>
>
> Thank you for your insightful review, you raise a valid point, and MLLMs bias is an issue we aim to address.
>
> - Our evaluation system utilizes a "Traditional Metrics + MLLM-based Assessment" approach. Traditional metrics act as a "Stability Anchor" to mitigate potential biases in MLLMs evaluations.
> - We performed rigorous human alignment experiments in **Sec. 4.4** and **Table 4**. The results demonstrate that our MLLM-based metrics maintain extremely high Spearman correlation coefficients with human ratings across tasks. This proves that the MLLMs' final scoring logic remains highly consistent with human perception, confirming its reliability as an evaluator.
> - To further reduce bias, we have added support in the code for switching between different MLLMs and using multiple MLLM models to average evaluation scores. This update will be included in the open-source code.
> - **Innovation:** We introduced MLLMs to address pain points specific to **Instruction-Guided Video Editing** that traditional metrics cannot resolve, such as complex spatial changes and semantic granularity. Our innovation in evaluation metrics is not merely the introduction of MLLMs, but a **3-Dimensional Evaluation Protocol** specifically designed for IVE characteristics, combining MLLM semantic understanding with traditional visual features.
>
>
>
> #### Q4: IVEBench Test Set Scale
>
> We sincerely appreciate your valuable review. Regarding the scale of the test set, we had to take into account both the labor costs and the computational costs associated with model evaluation. Specifically, the reasons are as follows:
>
> - The 600 videos are derived from 7 dimensions and 35 themes. Each video in the IVEBench dataset required multi-step manual screening and modification (including the video and matching prompts). These 600 videos were curated from a pool of **5,000+** videos, resulting in high annotation costs.
> - Regarding the benchmark dataset size, we had to consider the computational resources required for evaluation. We aimed to balance computational cost and time with distribution and diversity. Based on our tests using **dual H20 GPUs**, the average test time for a single video is **1.7 min - 2.5 min** (depending on the model's supported resolution and frames). Evaluating 600 videos on dual H20s takes approximately **20 hours**, which is a significant computational cost. It is also worth noting that before evaluation, the video editing model must generate the target videos using the original video and edit prompt, which adds a substantial computational burden. Therefore, we selected 600 videos to balance diversity and distribution against practically feasible computational costs.

---

> ### Author Response · Authors · 2025-11-27
> **[Paper 1859] Follow-up Inquiry regarding IVEbench**
>
> Dear Reviewer xwi2,
> I hope this message finds you well.
> As the discussion period is nearing its end with less than one week remaining, I wanted to ensure that we have addressed all your concerns satisfactorily. If there are any additional points or feedback you'd like us to consider, please let us know. Your insights are invaluable to us, and we are eager to address any remaining issues to improve our work.
> Thank you for your time and effort in reviewing our paper.

---

### Official Review · Reviewer_84vm · 2025-10-30

**Soundness:** 3
**Presentation:** 4
**Contribution:** 3
**Rating:** 6
**Confidence:** 4

**Summary:**

This paper introduces IVEBench, a new and modern benchmark designed to properly test today's instruction video editing models. The authors point out that existing benchmarks are falling behind; they use a limited variety of videos, cover only basic editing tasks (like changing style or swapping an object), and rely on outdated metrics that don't capture the full picture.

To tackle this, IVEBench brings three big upgrades. First, it features a diverse collection of 600 high-quality videos, spanning different themes, scenes, and lengths. Second, it includes a massive range of editing tasks: 8 categories with 35 sub-types, from simple attribute changes to complex camera movements—with prompts generated by LLMs and polished by experts. Finally, it introduces a smarter, three-dimensional evaluation system: Video Quality, Instruction Compliance, and Video Fidelity. Crucially, it leverages Multimodal LLMs to score complex edits that traditional metrics can't handle, ensuring the results are much closer to what a human would think.

**Strengths:**

1. A Massive and Diverse Dataset: Unlike older benchmarks with a handful of videos, IVEBench provides 600 clips, including both short and long ones (up to 1,024 frames!). This variety really pushes models to prove they can work on all kinds of content, not just cherry-picked examples.
2. Goes Beyond Basic Edits: The benchmark covers editing tasks that are specific to video, like adding camera motion or changing subject motion. This is a huge step up from benchmarks that just treat video as a series of images.
3. Smart, MLLM-Powered Evaluation: Using MLLMs to judge "Instruction Satisfaction" and "Content Fidelity". It allows the benchmark to automatically score tricky instructions (e.g., "change the camera to a high angle") that are almost impossible to measure with old-school pixel-based metrics.
4. Strong Alignment with Human Judgment: This work show that their automated metrics are highly correlated with human preferences (with Spearman's Rho scores often above 0.9). This means the benchmark isn't just spitting out numbers—it's reflecting what people actually see as a "good" or "bad" edit.

**Weaknesses:**

1. The initial evaluation only tests four models. While these are representative, the benchmark's true power will be seen when it's used to compare a much wider range of open-source and commercial models over time.
2. Potential Bias in MLLM-Based Evaluation: Relying on a specific MLLM (Qwen2.5-VL) for scoring could introduce subtle biases. For instance, if a new video editing model is built on a similar architecture to Qwen, the evaluator might unintentionally favor it. Diversifying the MLLM evaluators in the future could make the results even more robust.

**Questions:**

1.  The MLLM-based metrics are fantastic for evaluating complex instructions. But how do you plan to prevent the benchmark from becoming an "echo chamber," where editing models are trained to please a specific MLLM evaluator rather than genuine human users?

2. Edit instructions are often messy and combine multiple steps (e.g., "Make the cat look like it's made of fire and have it jump onto the table"). Does IVEBench have a plan to assess these kinds of compositional, multi-part edits in the future?

---

> ### Author Response · Authors · 2025-11-22
>
> #### Q1: Evaluated Models
>
> We strictly appreciate your insightful review. Due to the limited number of open-source works available at the time of submission, we initially evaluated only four models. We have been continuously monitoring works like InstructX [1r] and UniVideo [2r] regarding their open-source status.
>
> - We have kept track of the latest open-source IVE works and have now evaluated and added them to the test results (including **Lucy-Edit-Dev [3r], Omni-Video [4r], ICVE [5r], Ditto [6r]**). The specific results are shown in the table below, and the evaluation results and visualizations have been updated in **Table 2, Table 3, and Figure 4** of the paper.
>
> | **Database** | **Method**              | **Time per Frame ↓** | **Max Memory ↓** | **Video Resolution** |
> | ------------ | ----------------------- | -------------------- | ---------------- | -------------------- |
> | **Short**    | InsV2V                  | 3.96s                | **12.81GB**      | 512×512              |
> |              | AnyV2V                  | 11.66s               | 27.37GB          | 512×512              |
> |              | StableV2V               | 3.90s                | 28.31GB          | 512×512              |
> |              | VACE$^{\ddagger}$       | 27.03s               | 122.18GB         | **1280×720**         |
> |              | Lucy-Edit-Dev           | **1.52s**            | 32.21GB          | 832×480              |
> |              | Omni-Video$^{\ddagger}$ | 1.80s                | 36.37GB          | 640×352              |
> |              | ICVE$^{\ddagger}$       | 5.05s                | 88.33GB          | 384×240              |
> |              | Ditto$^{\ddagger}$      | 19.69s               | 38.49GB          | 832×480              |
> | **Long**     | InsV2V                  | 4.05s                | **13.48GB**      | 512×512              |
> |              | AnyV2V$^{\dagger}$      | 11.47s               | 63.15GB          | 512×512              |
> |              | StableV2V$^{\dagger}$   | 3.72s                | 49.82GB          | 512×512              |
> |              | VACE$^{\ddagger}$       | 51.00s               | 132.90GB         | **1280×720**         |
> |              | Lucy-Edit-Dev           | 2.23s                | 34.33GB          | 832×480              |
> |              | Omni-Video$^{\ddagger}$ | **2.04s**            | 36.37GB          | 640×352              |
> |              | ICVE$^{\ddagger}$       | 5.28s                | 91.67GB          | 384×240              |
> |              | Ditto$^{\ddagger}$      | 20.15s               | 38.86GB          | 832×480              |
>
> - From the numerical results in updated Table 2 and updated Figure 4, it can be observed that eight evaluated methods demonstrate relatively good frame-to-frame consistency. However, the per-frame image quality remains unsatisfactory, which consequently leads to low Video Fidelity scores. Moreover, these methods achieve very limited performance in instruction adherence, primarily due to the narrow range of task types they support. Among them, Lucy-Edit-Dev [3r] exhibits the best performance in editing speed. Ditto [6r] and InsV2V [7r] demonstrate the best performance in terms of editing capability, as they can execute editing prompts while preserving visual integrity. Furthermore, since Ditto successfully handles a larger number of cases, it achieves the best performance in Instruction Compliance. Nevertheless, these models achieve a Total Score of no more than 0.7 and an Instruction Compliance score of no more than 0.5, indicating that existing IVE methods still have substantial room for improvement in overall editing capability, particularly in Instruction Compliance.
> - **Commercial Models:** Considering cost limitations, we cannot currently support the use of commercial APIs. However, we are actively seeking support from supervisors and industry leaders. If we can secure access in the future, we will add evaluations for commercial models.

---

> > ### Author Response · Authors · 2025-11-22
> > **the updated Table 2**
> >
> > | **Database** | **Method**              | **Total Score** | **Video Quality** | **Instruction Compliance** | **Video Fidelity** | **SC**   | **BC**   | **TF**   | **MS**   | **VTSS**  | **OSC**  | **PSC**  | **IS**   | **QA**   | **SF**   | **MF**   | **CF**   |
> > | ------------ | ----------------------- | --------------- | ----------------- | -------------------------- | ------------------ | -------- | -------- | -------- | -------- | --------- | -------- | -------- | -------- | -------- | -------- | -------- | -------- |
> > | **Short**    | InsV2V                  | 0.67            | 0.80              | 0.39                       | 0.82               | 0.94     | 0.96     | 0.97     | 0.97     | 0.045     | 0.24     | 0.23     | 3.10     | 0.30     | 0.95     | 0.86     | **4.05** |
> > |              | AnyV2V                  | 0.58            | 0.73              | 0.42                       | 0.59               | 0.89     | 0.94     | 0.97     | 0.97     | 0.026     | 0.22     | **0.24** | 3.33     | 0.30     | 0.80     | 0.82     | 2.75     |
> > |              | StableV2V               | 0.51            | 0.69              | 0.43                       | 0.41               | 0.85     | 0.92     | 0.96     | 0.96     | 0.019     | 0.20     | **0.24** | 3.56     | 0.20     | 0.70     | 0.75     | 1.79     |
> > |              | VACE$^{\ddagger}$       | 0.63            | 0.80              | 0.25                       | **0.83**           | 0.95     | **0.98** | 0.98     | 0.98     | 0.045     | 0.23     | 0.22     | 2.16     | 0.20     | **0.97** | **0.89** | 4.03     |
> > |              | Lucy-Edit-Dev           | 0.64            | **0.82**          | 0.34                       | 0.75               | 0.95     | 0.96     | 0.98     | 0.99     | **0.051** | 0.24     | 0.22     | 2.84     | 0.20     | 0.93     | 0.68     | 3.83     |
> > |              | Omni-Video$^{\ddagger}$ | 0.59            | 0.78              | 0.44                       | 0.54               | **0.96** | 0.97     | 0.98     | 0.99     | 0.038     | 0.22     | 0.23     | 3.36     | **0.40** | 0.81     | 0.51     | 2.85     |
> > |              | ICVE$^{\ddagger}$       | 0.60            | 0.71              | 0.45                       | 0.64               | 0.95     | 0.97     | **0.99** | **1.00** | 0.017     | 0.23     | 0.23     | 3.62     | 0.30     | 0.85     | 0.46     | 3.55     |
> > |              | Ditto$^{\ddagger}$      | **0.67**        | 0.78              | **0.49**                   | 0.73               | **0.96** | **0.98** | 0.97     | 0.99     | 0.038     | **0.25** | **0.24** | **3.87** | 0.30     | 0.89     | 0.79     | 3.64     |
> > | **Long**     | InsV2V                  | 0.66            | 0.80              | 0.37                       | 0.79               | 0.90     | 0.94     | 0.98     | 0.98     | 0.048     | **0.24** | 0.23     | 3.10     | 0.20     | 0.95     | 0.68     | **4.13** |
> > |              | AnyV2V$^{\dagger}$      | 0.55            | 0.72              | 0.36                       | 0.57               | 0.84     | 0.92     | 0.97     | 0.97     | 0.029     | 0.22     | 0.23     | 3.25     | 0.00     | 0.80     | 0.82     | 2.65     |
> > |              | StableV2VE$^{\dagger}$  | 0.51            | 0.69              | 0.42                       | 0.41               | 0.83     | 0.91     | 0.96     | 0.96     | 0.021     | 0.23     | 0.23     | 3.45     | **0.25** | 0.70     | 0.77     | 1.79     |
> > |              | VACE$^{\ddagger}$       | 0.62            | 0.80              | 0.27                       | 0.78               | 0.92     | 0.95     | 0.96     | 0.96     | 0.048     | **0.24** | 0.22     | 2.27     | 0.20     | 0.96     | **0.96** | 3.74     |
> > |              | Lucy-Edit-Dev           | 0.65            | **0.82**          | 0.32                       | **0.81**           | 0.91     | 0.95     | 0.98     | 0.99     | **0.053** | **0.24** | 0.22     | 2.65     | 0.20     | **0.97** | 0.73     | 4.13     |
> > |              | Omni-Video$^{\ddagger}$ | 0.57            | 0.78              | 0.42                       | 0.51               | 0.94     | 0.96     | 0.97     | 0.98     | 0.039     | 0.22     | 0.23     | 3.53     | 0.20     | 0.81     | 0.55     | 2.59     |
> > |              | ICVE$^{\ddagger}$       | 0.59            | 0.72              | 0.40                       | 0.64               | 0.95     | 0.97     | **0.99** | **1.00** | 0.019     | 0.23     | 0.23     | 3.63     | 0.00     | 0.86     | 0.48     | 3.48     |
> > |              | Ditto$^{\ddagger}$      | **0.66**        | 0.78              | **0.48**                   | 0.72               | **0.96** | **0.98** | 0.97     | 0.99     | 0.038     | 0.23     | **0.24** | **3.93** | 0.20     | 0.86     | 0.73     | 3.69     |

---

> > > ### Author Response · Authors · 2025-11-22
> > >
> > > #### Q2: Potential Bias in MLLM Evaluation
> > >
> > > We appreciate your valuable review. You make a valid point, and the potential bias of MLLMs is an issue we are attempting to address.
> > >
> > > - Our evaluation system adopts a "Traditional Metrics + MLLM-based Assessment" approach. Traditional metrics serve as a "Stability Anchor" to mitigate potential biases in MLLM evaluations.
> > > - We conducted rigorous human alignment experiments in **Sec. 4.4** and **Table 4**. The results show that our MLLM-based metrics maintain extremely high Spearman correlation coefficients with human scores across various tasks. This proves that the final scoring logic of the MLLM remains highly consistent with human perception, making it a reliable evaluator.
> > > - We also fully agree with your view on introducing diverse MLLM evaluation models to improve stability. Therefore, we have supplemented the code to support **switching between different MLLMs** as well as **using multiple MLLM models to average the evaluation scores**. This will be updated in the open-source code.
> > >
> > >
> > >
> > > #### Q3: Avoiding Echo Chambers
> > >
> > > Thank you for your insightful review.
> > >
> > > - IVEBench is not a "black box" scorer that relies solely on MLLMs. It is a hybrid evaluation system composed of models with completely different architectures. Therefore, if a model attempts to "game" or "please" a specific MLLM, it will likely receive unsatisfactory scores on other independent metrics.
> > > - Our code has been updated to support evaluation using different MLLMs and averaging scores from multiple MLLMs. We will also maintain updates with the latest MLLMs, which can mitigate this issue to a certain extent.
> > > - We also recognize the value of using online **ELO ratings** to obtain more accurate user preference scores through real human choices. In the future, provided we have the resource support, we will consider using this method to supplement our evaluation.
> > >
> > >
> > >
> > > #### Q4: Complex Editing Instructions
> > >
> > >
> > >
> > > Thank you for your insightful review.
> > >
> > > - Firstly, the current IVEBench includes not only simple editing instructions but also composite instructions containing multiple requirements. An example of a composite multi-requirement instruction is from `short_0369`: *"Add a particle effect transition, then show the man wearing sunglasses and smiling."*
> > > - The current version of the IVEBench dataset does contain a large number of simple instructions. This is due to the nascent stage of video editing capabilities. Although the instructions are simple, current state-of-the-art open-source IVE methods still perform poorly on the metrics (specifically, evaluated open-source models score no higher than 0.5 on the **Instruction Compliance** dimension). If we were to focus primarily on complex instructions at this stage, the excessive difficulty would make it impossible to clearly compare the editing performance of different models.
> > > - Moving forward, we plan to expand IVEBench by launching a dataset focused on composite, multi-step editing instructions to adapt to the progress of future IVE methods and evaluation standards.
> > >
> > >
> > >
> > > [1r] Mou, Chong, et al. "InstructX: Towards unified visual editing with MLLM guidance." arXiv, 2025.
> > >
> > > [2r] Wei, Cong, et al. "UniVideo: Unified understanding, generation, and editing for videos." arXiv, 2025.
> > >
> > > [3r] DecartAI Team. "Lucy edit: High-fidelity text-guided video editing." Technical Report, 2025.
> > >
> > > [4r] Wang, Junke, et al. "Omni-Video: Democratizing unified video understanding and generation." arXiv, 2025.
> > >
> > > [5r] Liao, Xinyao, et al. "In-context learning with unpaired clips for instruction-based video editing." arXiv, 2025.
> > >
> > > [6r] Bai, Qingyan, et al. "Scaling instruction-based video editing with a high-quality synthetic dataset." arXiv, 2025.
> > >
> > > [7r] Cheng, Jiaxin, et al. "Consistent video-to-video transfer using synthetic dataset." arXiv, 2023.

---

> ### Author Response · Authors · 2025-11-27
> **[Paper 1859] Follow-up Inquiry regarding IVEbench**
>
> Dear Reviewer 84vm,
> I hope this message finds you well.
> As the discussion period is nearing its end with less than one week remaining, I wanted to ensure that we have addressed all your concerns satisfactorily. If there are any additional points or feedback you'd like us to consider, please let us know. Your insights are invaluable to us, and we are eager to address any remaining issues to improve our work.
> Thank you for your time and effort in reviewing our paper.

---

### Official Review · Reviewer_W9Es · 2025-11-01

**Soundness:** 3
**Presentation:** 4
**Contribution:** 3
**Rating:** 6
**Confidence:** 4

**Summary:**

This paper presents IVEBench, a benchmark for evaluating instruction-guided video editing. It includes 600 videos with diverse content and multiple editing categories, along with a three-dimensional evaluation protocol covering video quality, instruction compliance, and video fidelity. The authors also combine traditional metrics with multimodal large language model evaluations to better capture semantic alignment between edits and instructions.

**Strengths:**

The paper provides a well-structured benchmark in evaluating instruction-based video editing with diverse videos and a clear evaluation framework. The idea of combining automatic metrics with MLLM-based assessments is creative and reflects current research trends. The presentation is clear, the examples are easy to follow, and the benchmark has the potential to become a useful resource for future studies in this area.

**Weaknesses:**

I believe in video editing, the most crucial factor should be instruction compliance (whether the model actually does what the user asks for). From the qualitative examples, VACE achieve relatively high overall scores despite barely changing the video content (mostly just color tone adjustments). A model can still perform well even if it doesn’t really edit the content (even using the original video). This makes the benchmark a bit pointless. A weighted scoring system that puts more emphasis on instruction compliance would probably give a fairer picture of real editing performance.

**Questions:**

I just wanted to ask the authors if they’ve thought about reweighting the metrics so that instruction compliance has a bigger impact on the final score?

---

> ### Author Response · Authors · 2025-11-22
> **Rebuttal by Authors**
>
> We sincerely appreciate your valuable review; your insights are instrumental in refining IVEBench. While we acknowledge the critical importance of instruction adherence in video editing, our current weighting scheme (1:1:1) is empirically derived from the statistical feedback collected during our User Study (detailed in **Appendix C**). Our rationale and proposed solution are as follows:
>
> - **Rationale:** In human subjective evaluations, users exhibit an equally low tolerance for both **"editing failures"** and **"quality collapse."** In other words, a video that is *"correctly edited but suffers from severe semantic leakage"* is perceived as having similarly poor usability as a video that is *"barely edited but visually smooth."* Therefore, the 1:1:1 weighting objectively reflects the comprehensive expectations of users at the current technological stage. This validates the objective reality captured by IVEBench: stability remains a significant bottleneck in video editing.
> - **Solution:** Nevertheless, we fully agree that evaluation priorities may shift depending on specific application scenarios, distinctive needs, or future improvements in the stability of IVE models. Consequently, we will update our open-source evaluation code to **support easy customization of the weights for these three dimensions**.

---

> ### Author Response · Authors · 2025-11-27
> **[Paper 1859] Follow-up Inquiry regarding IVEbench**
>
> Dear Reviewer W9Es,
> I hope this message finds you well.
> As the discussion period is nearing its end with less than one week remaining, I wanted to ensure that we have addressed all your concerns satisfactorily. If there are any additional points or feedback you'd like us to consider, please let us know. Your insights are invaluable to us, and we are eager to address any remaining issues to improve our work.
> Thank you for your time and effort in reviewing our paper.

---

### Official Review · Reviewer_2aZn · 2025-11-01

**Soundness:** 2
**Presentation:** 3
**Contribution:** 3
**Rating:** 6
**Confidence:** 4

**Summary:**

This paper presents IVEBench, a modern benchmark suite specifically designed for Instruction-guided Video Editing (IVE), aiming to address the limitations of existing benchmarks, including insufficient diversity of video sources, narrow task coverage, and incomplete evaluation metrics. The suite contains 600 high-quality source videos covering 7 semantic dimensions, with frame lengths ranging from 32 to 1024 frames, divided into a short-sequence subset (400 videos) and a long-sequence subset (200 videos). It includes 8 major categories and 35 subcategories of editing tasks, generated by large language models (LLMs) and refined by experts.

IVEBench establishes a three-dimensional evaluation protocol—comprising Video Quality, Instruction Compliance, and Video Fidelity—integrating 12 metrics, including both traditional measures and MLLM-assisted assessments. In tests on four state-of-the-art IVE models, such as InsV2V and AnyV2V, IVEBench demonstrates high alignment with human perception (e.g., the VTSS metric achieves a Spearman correlation of 0.9982), and reveals existing model limitations in handling long videos (frame lengths exceeding 128 frames may cause memory overflow) and executing complex instructions (e.g., camera angle editing), providing a comprehensive standard for IVE evaluation.

**Strengths:**

1. **Comprehensive and well-constructed video and prompt design:**
   The paper collects a dataset of 600 videos across 7 dimensions, with each video paired with a corresponding prompt covering 8 different editing tasks. The dataset is large-scale and well-curated, with quality ensured through a combination of automatic preprocessing and manual screening. This provides a reliable foundation for evaluating video editing models.

2. **Thorough evaluation framework:**
   The study employs three primary evaluation dimensions and 12 fine-grained metrics, assessing video quality from multiple perspectives. This comprehensive approach ensures that different aspects of video editing performance are systematically measured.

3. **Provides insightful analysis:**
   Using the proposed evaluation framework, the authors analyze existing video editing models and reveal meaningful observations, such as high frame-to-frame consistency, weak single-frame quality, and limited support for diverse editing prompt types. These insights highlight both the strengths and current limitations of state-of-the-art video editing models.

**Weaknesses:**

1. **Ambiguity and overlap among evaluation dimensions:**
Although the paper proposes a comprehensive set of evaluation metrics, there exists noticeable overlap among certain dimensions. For example, both *Overall Semantic Consistency (OSC)* and *Instruction Satisfaction (IS)* aim to assess semantic alignment between the edited video and the given prompt, focusing on similar aspects of overall semantic coherence. This overlap raises concerns about the independence and discriminative validity of each metric.

2. **Limited reliability of human alignment experiments:**
The *human alignment* evaluation is conducted using only ten source videos, which is a relatively small sample size and may limit the representativeness of the results. Furthermore, the approach of directly using human rating distributions introduces subjectivity and potential bias, since human perceptual judgments are difficult to map linearly onto quantitative scales. The authors should clarify how they ensure consistency and statistical reliability in this human evaluation process.
3.**Limited diversity of evaluated models:**
The paper reports results from only four video editing models. This limited coverage weakens the generality of the analysis and makes it difficult to assess the broader applicability of the proposed benchmark. Evaluating more diverse or state-of-the-art models would significantly strengthen the conclusions.

**Questions:**

1. **On evaluation efficiency:**
   Given the large number of proposed evaluation metrics and assessment procedures, is there potential redundancy among them? Could the evaluation framework be simplified while maintaining its effectiveness? It would also be helpful to know the average time required to evaluate a single video under the current protocol.
2. **On the validity of the human rating process:**
How do the authors ensure that subjective human ratings align with objective quantitative metrics? Are there measures such as inter-rater reliability or calibration steps to minimize inconsistency among annotators?

---

> ### Author Response · Authors · 2025-11-22
> **Rebuttal by Authors**
>
> #### Q1: Metric Independence
>
> We appreciate your valuable reviews. Metric independence in a benchmark is crucial for avoiding redundancy and better revealing the trade-offs of the evaluated models. IVEBench's metrics demonstrate significant independence and distinctiveness for the following reasons:
>
> - **Different Types:** OSC is a traditional metric, whereas IS is an MLLM-based metric. MLLM-based metrics complement traditional metrics by capturing high-level semantics and granularity, such as complex semantic combinations and dynamic temporal changes. Conversely, traditional metrics serve as a "Stability Anchor" to mitigate potential biases in MLLM evaluations.
> - **Different Inputs:** The OSC metric takes the *target video* and *target prompt* as input, focusing on the overall content after editing. The PSC metric takes the *target video* and *target phrase*, focusing on the specific edited subject. The IS metric takes the *target video* and *edit prompt*, focusing specifically on the execution of the editing instruction.
> - **Quantitative Verification:** To quantitatively verify metric independence, we calculated the Spearman correlation matrix across all model results. The visualized heatmap is shown in **Figure A3**. The results show a correlation coefficient of only **0.16** between OSC and IS, and the Spearman correlation between metrics does not exceed **0.65** [1r]. This confirms the absence of multicollinearity between metrics and supports the independence and distinctiveness of our design. Furthermore, negative correlations exist between several metrics, indicating that IVEBench metrics effectively reveal the trade-offs of the evaluated models.
>
>
>
> #### Q2: Evaluation Time
>
> Thank you for your question; evaluation speed is indeed a critical attribute of a benchmark.
>
> - **Testing:** Based on our tests using dual H20 GPUs, the average testing time for a single video on IVEBench is **1.7 min - 2.5 min** (average testing time is influenced by the maximum frame count and resolution supported by the evaluated model).
> - **Update:** We will add automatic calculation of inference time to the codebase and update this in the open-source release.
>
>
>
> #### Q3: Reliability of Human Evaluation
>
> We appreciate your valuable review regarding the number of source videos used in human evaluation, as well as the consistency and statistical reliability of the process.
>
> - **Source Quantity:** The number of source videos for human alignment in IVEBench initially followed the approach of EditBoard [n], selecting 10 source videos and their corresponding prompts.
> - **Improvement:** We acknowledge the reviewer's suggestion. To enhance credibility while considering evaluation costs, we increased the number of source videos to **30** (the new set covers all editing categories and maximizes semantic themes). The comparison table of Spearman's $\rho$ below shows that the new results are consistent with the original ones, demonstrating that our metrics are well-aligned with human judgment.
>
> | **Metric**              | **SC** | **BC** | **TF** | **MS** | **VTSS** | **OSC** | **PSC** | **IS** | **QA** | **SF** | **MF** | **CF** |
> | ----------------------- | ------ | ------ | ------ | ------ | -------- | ------- | ------- | ------ | ------ | ------ | ------ | ------ |
> | **$\rho_{\text{new}}$** | 0.9536 | 0.9503 | 0.8842 | 0.9774 | 0.9985   | 0.7210  | 0.8316  | 0.9834 | 0.8104 | 0.9453 | 0.9371 | 0.9892 |
> | **$\rho_{\text{old}}$** | 0.9583 | 0.9442 | 0.8907 | 0.9763 | 0.9982   | 0.7105  | 0.8465  | 0.9859 | 0.8216 | 0.9400 | 0.9373 | 0.9896 |
>
> - **Calibration:** We invited professional testers (with domain knowledge) and informed them of the meaning of each metric beforehand, providing positive and negative examples. We then conducted an additional pre-annotation test to ensure they fully understood the metrics. Testers were instructed to focus strictly on the specified metric and ignore overall video quality, as detailed in **Appendix F**.
> - **Inter-rater Reliability:** Acknowledging that individuals vary in evaluation, we calculated **Fleiss' Kappa** for our human annotations and obtained a score of **0.78**, which indicates **'substantial agreement'** according to Landis and Koch [2r]. This demonstrates that our rigorous calibration process effectively aligned the standards of different annotators.

---

> > ### Author Response · Authors · 2025-11-22
> > **Rebuttal by Authors**
> >
> > #### Q4: Evaluated Models
> >
> > We strictly appreciate your valuable review. Due to the limited number of open-source works available at the time of submission, we initially evaluated only four models. We have been continuously monitoring works like InstructX [3r] and UniVideo [4r] regarding their open-source status.
> >
> > - We have kept track of the latest open-source IVE works and have now evaluated and added them to the test results (including **Lucy-Edit-Dev [5r], Omni-Video [6r], ICVE [7r], Ditto [8r]**). The evaluation results and visualizations have been updated in **Table 2, Table 3, and Figure 4** of the paper.
> >
> > | **Database** | **Method**              | **Time per Frame ↓** | **Max Memory ↓** | **Video Resolution** |
> > | ------------ | ----------------------- | -------------------- | ---------------- | -------------------- |
> > | **Short**    | InsV2V                  | 3.96s                | **12.81GB**      | 512×512              |
> > |              | AnyV2V                  | 11.66s               | 27.37GB          | 512×512              |
> > |              | StableV2V               | 3.90s                | 28.31GB          | 512×512              |
> > |              | VACE$^{\ddagger}$       | 27.03s               | 122.18GB         | **1280×720**         |
> > |              | Lucy-Edit-Dev           | **1.52s**            | 32.21GB          | 832×480              |
> > |              | Omni-Video$^{\ddagger}$ | 1.80s                | 36.37GB          | 640×352              |
> > |              | ICVE$^{\ddagger}$       | 5.05s                | 88.33GB          | 384×240              |
> > |              | Ditto$^{\ddagger}$      | 19.69s               | 38.49GB          | 832×480              |
> > | **Long**     | InsV2V                  | 4.05s                | **13.48GB**      | 512×512              |
> > |              | AnyV2V$^{\dagger}$      | 11.47s               | 63.15GB          | 512×512              |
> > |              | StableV2V$^{\dagger}$   | 3.72s                | 49.82GB          | 512×512              |
> > |              | VACE$^{\ddagger}$       | 51.00s               | 132.90GB         | **1280×720**         |
> > |              | Lucy-Edit-Dev           | 2.23s                | 34.33GB          | 832×480              |
> > |              | Omni-Video$^{\ddagger}$ | **2.04s**            | 36.37GB          | 640×352              |
> > |              | ICVE$^{\ddagger}$       | 5.28s                | 91.67GB          | 384×240              |
> > |              | Ditto$^{\ddagger}$      | 20.15s               | 38.86GB          | 832×480              |
> >
> > - From the numerical results in updated Table 2 and updated Figure 4, it can be observed that eight evaluated methods demonstrate relatively good frame-to-frame consistency. However, the per-frame image quality remains unsatisfactory, which consequently leads to low Video Fidelity scores. Moreover, these methods achieve very limited performance in instruction adherence, primarily due to the narrow range of task types they support. Among them, Lucy-Edit-Dev [5r] exhibits the best performance in editing speed. Ditto [8r] and InsV2V [9r] demonstrate the best performance in terms of editing capability, as they can execute editing prompts while preserving visual integrity. Furthermore, since Ditto successfully handles a larger number of cases, it achieves the best performance in Instruction Compliance. Nevertheless, these models achieve a Total Score of no more than 0.7 and an Instruction Compliance score of no more than 0.5, indicating that existing IVE methods still have substantial room for improvement in overall editing capability, particularly in Instruction Compliance.
> >
> > [1r] Dormann, Carsten F., et al. "Collinearity: a review of methods to deal with it and a simulation study evaluating their performance." Ecography, 2013.
> >
> > [2r] Landis, J. Richard, et al. "The measurement of observer agreement for categorical data." Biometrics, 1977.
> >
> > [3r] Mou, Chong, et al. "InstructX: Towards unified visual editing with MLLM guidance." arXiv, 2025.
> >
> > [4r] Wei, Cong, et al. "UniVideo: Unified understanding, generation, and editing for videos." arXiv, 2025.
> >
> > [5r] DecartAI Team. "Lucy edit: High-fidelity text-guided video editing." Technical Report, 2025.
> >
> > [6r] Wang, Junke, et al. "Omni-Video: Democratizing unified video understanding and generation." arXiv, 2025.
> >
> > [7r] Liao, Xinyao, et al. "In-context learning with unpaired clips for instruction-based video editing." arXiv, 2025.
> >
> > [8r] Bai, Qingyan, et al. "Scaling instruction-based video editing with a high-quality synthetic dataset." arXiv, 2025.
> >
> > [9r] Cheng, Jiaxin, et al. "Consistent video-to-video transfer using synthetic dataset." arXiv, 2023.

---

> > > ### Author Response · Authors · 2025-11-22
> > > **the updated Table 2**
> > >
> > > | **Database** | **Method**              | **Total Score** | **Video Quality** | **Instruction Compliance** | **Video Fidelity** | **SC**   | **BC**   | **TF**   | **MS**   | **VTSS**  | **OSC**  | **PSC**  | **IS**   | **QA**   | **SF**   | **MF**   | **CF**   |
> > > | ------------ | ----------------------- | --------------- | ----------------- | -------------------------- | ------------------ | -------- | -------- | -------- | -------- | --------- | -------- | -------- | -------- | -------- | -------- | -------- | -------- |
> > > | **Short**    | InsV2V                  | 0.67            | 0.80              | 0.39                       | 0.82               | 0.94     | 0.96     | 0.97     | 0.97     | 0.045     | 0.24     | 0.23     | 3.10     | 0.30     | 0.95     | 0.86     | **4.05** |
> > > |              | AnyV2V                  | 0.58            | 0.73              | 0.42                       | 0.59               | 0.89     | 0.94     | 0.97     | 0.97     | 0.026     | 0.22     | **0.24** | 3.33     | 0.30     | 0.80     | 0.82     | 2.75     |
> > > |              | StableV2V               | 0.51            | 0.69              | 0.43                       | 0.41               | 0.85     | 0.92     | 0.96     | 0.96     | 0.019     | 0.20     | **0.24** | 3.56     | 0.20     | 0.70     | 0.75     | 1.79     |
> > > |              | VACE$^{\ddagger}$       | 0.63            | 0.80              | 0.25                       | **0.83**           | 0.95     | **0.98** | 0.98     | 0.98     | 0.045     | 0.23     | 0.22     | 2.16     | 0.20     | **0.97** | **0.89** | 4.03     |
> > > |              | Lucy-Edit-Dev           | 0.64            | **0.82**          | 0.34                       | 0.75               | 0.95     | 0.96     | 0.98     | 0.99     | **0.051** | 0.24     | 0.22     | 2.84     | 0.20     | 0.93     | 0.68     | 3.83     |
> > > |              | Omni-Video$^{\ddagger}$ | 0.59            | 0.78              | 0.44                       | 0.54               | **0.96** | 0.97     | 0.98     | 0.99     | 0.038     | 0.22     | 0.23     | 3.36     | **0.40** | 0.81     | 0.51     | 2.85     |
> > > |              | ICVE$^{\ddagger}$       | 0.60            | 0.71              | 0.45                       | 0.64               | 0.95     | 0.97     | **0.99** | **1.00** | 0.017     | 0.23     | 0.23     | 3.62     | 0.30     | 0.85     | 0.46     | 3.55     |
> > > |              | Ditto$^{\ddagger}$      | **0.67**        | 0.78              | **0.49**                   | 0.73               | **0.96** | **0.98** | 0.97     | 0.99     | 0.038     | **0.25** | **0.24** | **3.87** | 0.30     | 0.89     | 0.79     | 3.64     |
> > > | **Long**     | InsV2V                  | 0.66            | 0.80              | 0.37                       | 0.79               | 0.90     | 0.94     | 0.98     | 0.98     | 0.048     | **0.24** | 0.23     | 3.10     | 0.20     | 0.95     | 0.68     | **4.13** |
> > > |              | AnyV2V$^{\dagger}$      | 0.55            | 0.72              | 0.36                       | 0.57               | 0.84     | 0.92     | 0.97     | 0.97     | 0.029     | 0.22     | 0.23     | 3.25     | 0.00     | 0.80     | 0.82     | 2.65     |
> > > |              | StableV2VE$^{\dagger}$  | 0.51            | 0.69              | 0.42                       | 0.41               | 0.83     | 0.91     | 0.96     | 0.96     | 0.021     | 0.23     | 0.23     | 3.45     | **0.25** | 0.70     | 0.77     | 1.79     |
> > > |              | VACE$^{\ddagger}$       | 0.62            | 0.80              | 0.27                       | 0.78               | 0.92     | 0.95     | 0.96     | 0.96     | 0.048     | **0.24** | 0.22     | 2.27     | 0.20     | 0.96     | **0.96** | 3.74     |
> > > |              | Lucy-Edit-Dev           | 0.65            | **0.82**          | 0.32                       | **0.81**           | 0.91     | 0.95     | 0.98     | 0.99     | **0.053** | **0.24** | 0.22     | 2.65     | 0.20     | **0.97** | 0.73     | 4.13     |
> > > |              | Omni-Video$^{\ddagger}$ | 0.57            | 0.78              | 0.42                       | 0.51               | 0.94     | 0.96     | 0.97     | 0.98     | 0.039     | 0.22     | 0.23     | 3.53     | 0.20     | 0.81     | 0.55     | 2.59     |
> > > |              | ICVE$^{\ddagger}$       | 0.59            | 0.72              | 0.40                       | 0.64               | 0.95     | 0.97     | **0.99** | **1.00** | 0.019     | 0.23     | 0.23     | 3.63     | 0.00     | 0.86     | 0.48     | 3.48     |
> > > |              | Ditto$^{\ddagger}$      | **0.66**        | 0.78              | **0.48**                   | 0.72               | **0.96** | **0.98** | 0.97     | 0.99     | 0.038     | 0.23     | **0.24** | **3.93** | 0.20     | 0.86     | 0.73     | 3.69     |

---

> ### Author Response · Authors · 2025-11-27
> **[Paper 1859] Follow-up Inquiry regarding IVEbench**
>
> Dear Reviewer 2aZn,
> I hope this message finds you well.
> As the discussion period is nearing its end with less than one week remaining, I wanted to ensure that we have addressed all your concerns satisfactorily. If there are any additional points or feedback you'd like us to consider, please let us know. Your insights are invaluable to us, and we are eager to address any remaining issues to improve our work.
> Thank you for your time and effort in reviewing our paper.

---

### Author Response · Authors · 2025-11-22
**General Responses and Revision Summary**

**Dear All Reviewers:**

We sincerely thank all the reviewers for their constructive feedback and valuable insights. All these invaluable insights have significantly helped in making IVEBench more comprehensive and valuable.

We appreciate your thoughtful questions and concerns regarding evaluated model coverage, metric independence, and evaluation flexibility. We have tried our best to address them comprehensively in our individual responses.

We have carefully revised the paper to reflect our responses to reviewers' concerns and suggestions (highlighted in deepred). Key revisions and **future commitments regarding the codebase** are:

- **Expanded Model Evaluation:** We have evaluated and added **4 latest open-source IVE models** (Lucy-Edit-Dev, Omni-Video, ICVE, and Ditto) to the benchmark results. The corresponding **Table 2, Table 3, and Figure 4** have been updated to provide a more comprehensive comparison of the state-of-the-art.

- **Enhanced Human Evaluation Reliability:** We increased the number of source videos for human alignment from 10 to **30** and calculated **Fleiss' Kappa (0.78)**, quantitatively proving the substantial agreement and statistical reliability of our human evaluation process.

- **Verification of Metric Independence:** To address concerns about metric redundancy, we calculated and visualized the **Spearman correlation matrix** (Figure A3), quantitatively confirming the independence and distinctiveness of our designed metrics.

- **Mitigation of MLLM Bias (Code Update):** We **will update** the code to support **switching between different MLLMs** and **averaging scores from multiple MLLMs**, reducing potential bias from a single evaluator model.

- **Customizable Evaluation Weights (Code Update):** We **will update** the code to support **user-customizable weights** for the three evaluation dimensions (Video Quality, Instruction Compliance, Video Fidelity), allowing adaptability to different application scenarios.

- **Inference Time Calculation (Code Update):** We **will add** automatic calculation of inference time to the codebase in the upcoming open-source release to better assess model efficiency.

We are truly grateful once again for your feedback, which has significantly strengthened IVEBench. Please refer to our individual responses for our detailed answers to specific questions and concerns. We would be delighted to discuss them further and will spare no effort in addressing any remaining points.

Thank you, **Authors**

---

### Meta-Review · Area_Chair_tQXS · 2026-01-03

**Summary:**

The paper proposes IVEBench, a benchmark for instruction-guided video editing. It comprises a dataset of 600 source videos. The evaluation protocol consists of three-dimensional metrics, including video quality, instruction compliance, and video fidelity.

The reviewers agree that the proposed benchmark is comprehensive and well-constructed. The paper also provides insightful analysis. The benchmark can benefit future research in the area of video editing.

There are also several weaknesses pointed out by the reviewers, which have been addressed during the rebuttal phase. Therefore, I would recommend acceptance of this work. I encourage the authors to incorporate reviewers' suggestions in their next version.

**Reviewer Concerns:**

Most of the major concerns are addressed during the rebuttal phase. It's understandable that it's expensive to include commercial models in the evaluation, but it will be a nice thing to add in the future version.

**Reviewer Scores:**

6 6 6 4 -> 6 6 6 4

---

### Decision · Program_Chairs · 2026-01-26

Accept (Poster)